# Peritoneal tissue-resident macrophages are metabolically poised to engage microbes using tissue-niche fuels

Luke C. Davies [1,2], Christopher M. Rice[2], Erika M. Palmieri[2], Philip R. Taylor [1], Douglas B. Kuhns[3] & Daniel W. McVicar[2]

The importance of metabolism in macrophage function has been reported, but the in vivo relevance of the in vitro observations is still unclear. Here we show that macrophage metabolites are defined in a specific tissue context, and these metabolites are crucially linked to tissue-resident macrophage functions. We find the peritoneum to be rich in glutamate, a glutaminolysis-fuel that is exploited by peritoneal-resident macrophages to maintain respiratory burst during phagocytosis via enhancing mitochondrial complex-II metabolism. This niche-supported, inducible mitochondrial function is dependent on protein kinase C activity, and is required to fine-tune the cytokine responses that control inflammation. In addition, we find that peritoneal-resident macrophage mitochondria are recruited to phagosomes and produce mitochondrially derived reactive oxygen species, which are necessary for microbial killing. We propose that tissue-resident macrophages are metabolically poised in situ to protect and exploit their tissue-niche by utilising locally available fuels to implement specific metabolic programmes upon microbial sensing.

[1] Division of Infection & Immunity, School of Medicine, Cardiff University, Tenovus Building, Heath Park CF14 4XN, UK. [2] Cancer & Inflammation Program, National Cancer Institute, Frederick, MD 21702, USA. [3] Leidos Biomedical Research Inc., Frederick National Laboratory for Cancer Research, Frederick, MD 21702, USA. Correspondence and requests for materials should be addressed to L.C.D. (email: davieslc6@cf.ac.uk) or to D.W.M. (email: mcvicard@mail.nih.gov)

Tissue-resident macrophages (TRMØ) are tissue-specialised immune sentinels, which have key functions in homoeostasis and inflammation[1, 2]. Like many TRMØ, peritoneal TRMØ (pTRMØ) are not originally derived from monocytes, but rather from embryonic progenitors seeded into tissues before birth, with the populations maintained by local proliferation[3–6]. These cells exist in complex environments and do not fit traditional polarisation categories, such as LPS and interferon-γ stimulated pro-inflammatory (M1) and interleukin-4 stimulated anti-inflammatory (M2)[2, 7]. It has been shown that tissue-niche environments can govern cell phenotype via epigenetic programming[8, 9]. The precise factors responsible for this in situ programming are largely uncategorised, although peritoneal retinoic acid can induce *Gata6* expression[10] to dictate pTRMØ phenotype[10–12]. However, the metabolic repertoire of in situ environments are largely unknown, and it is likely that other metabolites will govern resident cell functions in their respective tissues[2].

There has been resurgent interest in metabolic control of cellular function, particularly in immunology[13–15]. In bone marrow-derived macrophages (BMDM), M1 macrophage differentiation in vitro and functions, including cytokine production, are dependent on glucose and glutamine metabolism, whereas the M2 phenotype reportedly relies on fatty acid oxidation for oxidative phosphorylation (OXPHOS), and on glutamine for protein modifications[15, 16]. However, M2 differentiation itself does not require long-chain fatty acid oxidation[17, 18]. By contrast, little is known about TRMØ metabolism, or which fuels are available in tissue-niches that govern cell function[2].

Macrophage biology encompasses common and tissue-niche functions. For example, pTRMØ invade peritoneal organs and facilitate repair[19], supporting their physiological importance. However, TRMØ are also the front-line immune sentinels that possess primary macrophage functions including phagocytosis and respiratory burst[20]. TRMØ express a repertoire of pattern-recognition receptors, including Dectin-1[21], mannose receptor and multiple toll-like receptors. These receptors engage microbes to promote phagocytosis and assembly of NADPH oxidase (NOX) 2 that supports oxidative respiratory burst[22]. Although respiratory burst and the related signalling mechanisms are understood in neutrophils, the metabolic requirements of respiratory burst are not clear in TRMØ.

Here our results link the availability of peritoneal fuels and requirements for pTRMØ metabolic processes that sustain

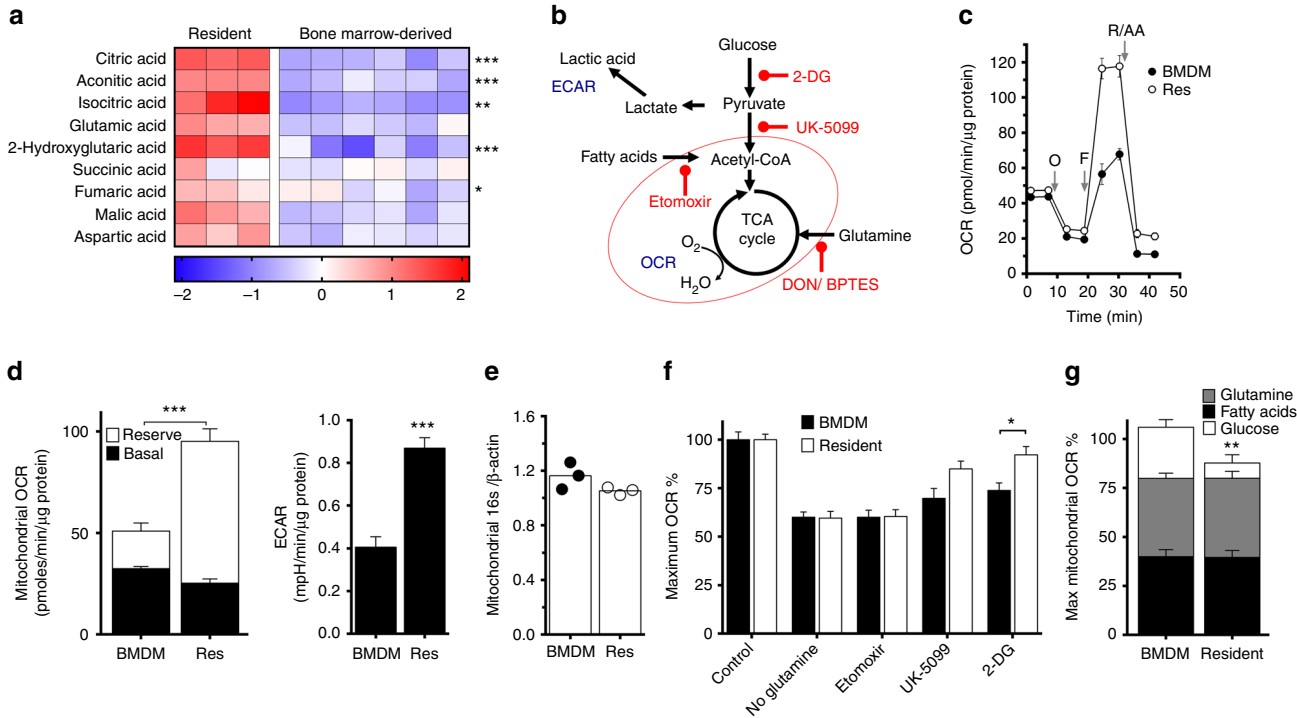

**Fig. 1** Peritoneal tissue-resident macrophages are highly oxidative and are fuelled by glutamine. **a** Heat-map showing the log10 difference from the average metabolite peak areas from gas chromatography-mass spectrometry, which are associated with the citric acid cycle from peritoneal tissue-resident macrophages (pTRMØ, Res) and bone marrow-derived macrophages (BMDM). Data show 3–6 independent samples per group analysed by two-way ANOVA (interaction < 0.0001) with Sidak's post-tests. **b** Schematic showing a simplified metabolic pathway detailing readouts for extracellular acidification rate (ECAR), oxygen consumption rate (OCR) and drugs, which inhibit these pathways, the red oval indicates the mitochondria. **c** Graph showing representative Seahorse mitochondrial stress tests, O = oligomycin (1.26 μM), F = FCCP (0.67 μM), R/AA = Rotenone (0.2 μM) + antimycin A (1 μM). **d** Bar graphs showing quantified protein-normalised ECAR and mitochondrial OCR from stress tests in **c**, combination of the two OCR parameters represents the maximum mitochondrial capacity. Data (n = 3 separate wells per group) represent three experiments, OCR data were analysed by paired two-way ANOVA (interaction p < 0.0001) with Sidak's post-tests for the difference between pTRMØ and BMDM (reserve/ maximal mitochondrial capacity) or by Student's t-test (ECAR). **e** Bar graph depicting the ratio of mitochondrial 16s deoxyribonucleic acid (DNA) vs β-actin DNA in three independent samples, data was not significant by Student's t-test. **f** Bar graph showing the drop in maximum decoupled mitochondrial OCR after addition of drugs or withdrawal of fuels. Glucose dependence was assessed using 2-deoxyglucose (2-DG, 100 mM), pyruvate using UK-5099 (20μM), a mitochondrial pyruvate transport inhibitor, long-chain fatty acids by etomoxir (100 μM) and glutamine by 1 h starvation before reintroduction of glutamine (2 mM) or control media. **g** Stacked bar graph showing a breakdown of the effects of different metabolic fuels on maximum mitochondrial capacity from **f**. Data **f** and **g** (n = at least 3 separate wells per group) represent at least two experiments, and were analysed by two-way ANOVA ((G) interaction p = 0.02; (F) interaction p = 0.056, cell p = 0.017, fuel p < 0.0001) with Sidak's post-tests. All error bars denote mean ± SEM

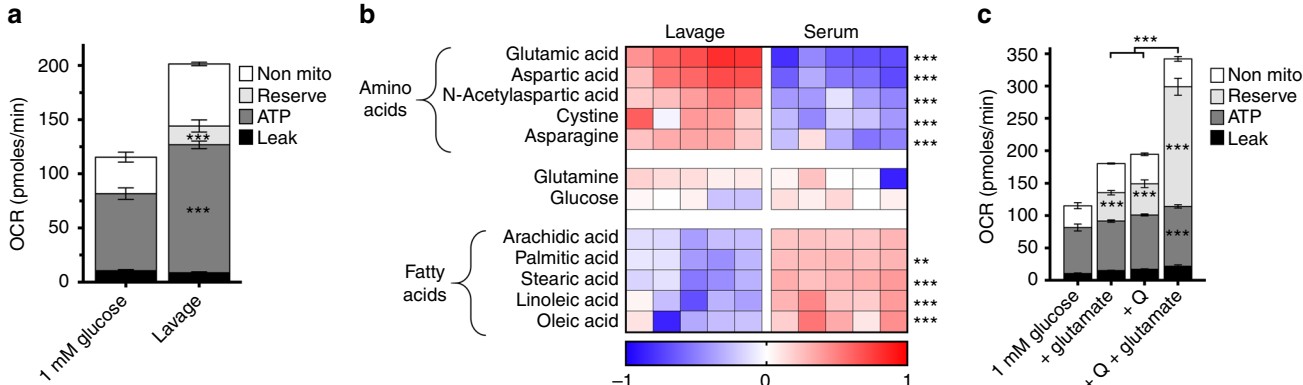

**Fig. 2** Peritoneal tissue-resident macrophage mitochondria can be fuelled through abundant peritoneal glutamate. **a** Bar graph showing the effect of 5× diluted peritoneal fluid (1 mM glucose media lavage followed by cell and protein depletion) in comparison to 1 mM glucose media on mitochondrial parameters in peritoneal tissue-resident macrophages. Data were from one experiment ($n = 5$–6 separate wells per group) and analysed by paired two-way ANOVA (interaction $p < 0.0001$) with Sidak's post-tests. **b** Heat-map showing the log10 difference from the average metabolite peak areas from gas chromatography-mass spectrometry analysis of the most enriched amino acids in peritoneal lavage fluid and the fatty acids in blood serum, glucose and glutamine act as a reference. Data show five independent samples that were analysed by two-way ANOVA (interaction $p < 0.0001$) with Sidak's post-tests. **c** Stacked graph detailing the addition of glutamine (Q, 0.25 mM) and glutamate (0.5 mM) to peritoneal tissue-resident macrophages and resulting changes in mitochondrial parameters. Data are from the same experiment as **a**, but increases in mitochondrial function after glutamate and Q represent three independent experiments, data were analysed by two-way ANOVA (interaction $p < 0.0001$), Sidak's post-tests are indicated within the bars for parameters vs 1 mM glucose and above representing the synergistic effect of glutamate and Q. All error bars denote mean ± SEM

respiratory burst. Resting pTRMØ are resistant to nutrient depletion and have very little basal mitochondrial complex-II (CII) activity. However, upon phagocytosis or metabolic stress, pTRMØ utilise peritoneal metabolites to promote an enhancement in glutaminolysis-fuelled CII metabolism that facilitates respiratory burst required for microbial control and immune function.

## Results

**PTRMØ have a substantial glutamate-fuelled mitochondrial reserve.** The majority of MØ research has been performed using cultured cell lines and/or in vitro-derived BMDM. Accordingly, precise mechanisms of how cells meet metabolic demands in vivo remain unknown, making it critical to assess cell function in conjunction with tissue-niche metabolites. The peritoneum contains pTRMØ that are readily recovered without disruptive tissue digestion, limiting alterations of in vivo-characteristics. We performed gas chromatography-time-of-flight metabolite analysis to determine metabolic differences between macrophage colony stimulating factor (M-CSF)-derived BMDM, the standard model for macrophage functional analysis, and pTRMØ. Analysis revealed a stark contrast, with many metabolites having statistically significant differences (Two-way ANOVA post-tests $p < 0.05$) (Supplementary Fig. 1; Supplementary Data 1); however, unbiased ingenuity pathway analysis (IPA) described enrichment of tri-carboxylic acid (TCA) cycle-associated metabolites (Fig. 1a) and lower amino acids in pTRMØ compared to BMDM (Supplementary Fig. 1). Given these differences, we next compared their RNA expression. RNA-sequencing revealed expected contrasts in gene expression, although surprisingly, IPA analysis did not identify metabolic pathways as differentially regulated (Supplementary Fig. 2). However, there were statistically significant differences in many metabolic genes (Limma-Voom-paired tests, $p < 0.05$). This involved alternate expression of genes within the same metabolic pathways, including different isoform expression between the populations (Supplementary Fig. 3); explaining the lack of IPA identification. Together, these data support that pTRMØ may have greater mitochondrial activity than BMDM and perhaps deplete amino acids to support this. To assess

mitochondrial function, we quantified oxygen consumption rates (OCR) during mitochondrial stress tests (Fig. 1b, c). Interestingly, basal-OCR was comparable, however maximal mitochondrial capacity was greater in pTRMØ (Fig. 1c, d), despite similar mitochondrial quantification (Fig. 1e). Accordingly, we analysed mitochondrial volume in individual MØs (Supplementary Movies 1–4). Mitochondrial morphology, volumes and numbers were heterogeneous (Supplementary Fig. 4a), although on average pTRMØ had fewer, but larger mitochondrial networks (Supplementary Fig. 4b), resulting in similar total mitochondrial volumes (Supplementary Fig. 4c). Collectively, this demonstrates that pTRMØ have similar mitochondrial volume but higher OXPHOS during metabolic stress, indicating greater mitochondrial efficiency[23]. Furthermore, pTRMØ exhibited higher extracellular acidification rates (ECAR), an indicator of glycolytic activity (Fig. 1b, d).

OXPHOS is primarily fuelled from three sources: fatty acid oxidation, glycolysis or glutaminolysis. We sought to establish which fuels were used in maintenance of MØ mitochondrial function using inhibitors and fuel deprivation (Fig. 1b). Basal-OCRs of pTRMØ were resistant to nutrient stress, previously attributed to an internal pool of stored metabolites[24]. However, upon mitochondrial decoupling, the dependence of maximal OCR on metabolic pathways could be interrogated. Despite different origins, BMDM and pTRMØ display similar fuel dependence, although pTRMØ required less glucose-derived pyruvate (Fig. 1f, g). Together, these data demonstrate that pTRMØ have approximately double the mitochondrial capacity of BMDM, which is primarily dependent on glutamine and fatty acids.

Subsequently, we investigated whether peritoneal metabolites can support pTRMØ mitochondria. Indeed, peritoneal fluid augmented maximal OCR (Fig. 2a), demonstrating that peritoneal fuels support pTRMØ mitochondrial function. To identify specific fuels, we performed gas chromatography-time-of-flight metabolomic analysis on peritoneal fluid. Serum supplies metabolic fuels (such as glucose) from digestion to the body including the peritoneum. Therefore, to test whether specific metabolites might be enriched in the peritoneum compared to

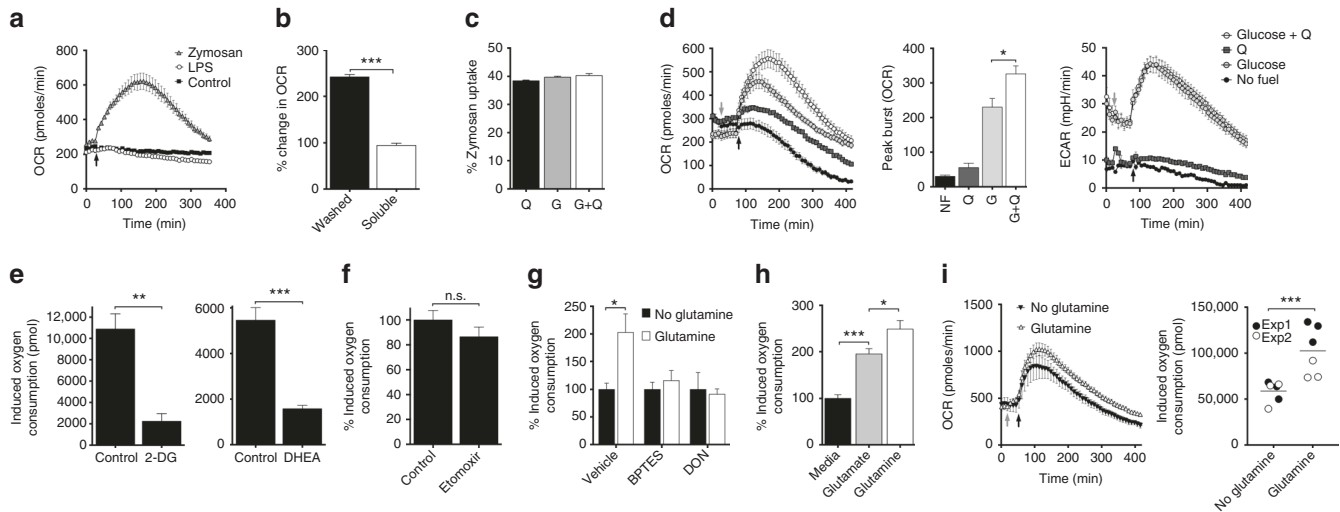

**Fig. 3** Peritoneal tissue-resident macrophages enhance their respiratory burst with glutaminolysis. **a** Line graph showing oxygen consumption rate (OCR) in peritoneal tissue-resident macrophages (pTRMØ). Arrow = zymosan (50 μg/ml), LPS (100 ng/ml) or control. **b** OCR changes in pTRMØ with washed particle- or soluble-zymosan components. Data were analysed by Student's *t*-test. **c** Quantification of pTRMØ uptake of pHrodo zymosan (50 μg/ml,1 h). G = glucose (25 mM), Q = glutamine (2 mM). Data were not significant by one-way ANOVA (*p* = 0.0756) with Tukey's post-tests. **d** OCR or extracellular acidification rate (ECAR) vs time in pTRMØ. Cells were starved from glutamine (Q) and cultured ± glucose before addition of Q or control (grey arrow). Zymosan = black arrow. Peak burst is quantified in the chart and analysed by one-way ANOVA (*p* < 0.0001), significant Tukey's post-tests for Q addition are shown. **e** Graph showing the total oxygen consumption of pTRMØ after zymosan, quantified from OCR vs time (as in **d**), in the presence of 2-deoxyglucose (2-DG, 100 mM), dehydroepiandrosterone (DHEA,100 μM) or vehicle. Data were analysed by Student's *t*-test. **f** Graph showing the percentage oxygen consumption of pTRMØ after zymosan in the presence of etomoxir (50 μM) or vehicle. Data are pool of two experiments (*n* = 6–7) and were analysed by two-way ANOVA (interaction *p* = 0.376, etomoxir *p* = 0.2588). **g** Relative total oxygen consumption after zymosan of pTRMØ pre-treated with glutaminolysis inhibitors bis-phenylacetamido-thiadiazolyl-ethyl sulphide (BPTES, 15 μM) and diazo-oxo-norleucine (DON, 10 mM), in the presence or absence of Q. Data were analysed by two-way ANOVA (interaction *p* = 0.0536) with Sidak's post-tests. **h** Total oxygen consumption after zymosan of pTRMØ in the presence or absence of Q (4 mM) or glutamate (4 mM). Data were analysed by one-way ANOVA (*p* < 0.0001) with Tukey's post-tests. **i** Human monocytes were pre-cultured for 48 h with recombinant human M-CSF (50 ng/ml) and interferon-γ (10 ng/ml). Oxygen consumption was quantified as in **d** and shows six separate wells per group pooled from two experiments, analysed by two-way ANOVA (interaction *p* = 0.1231), the *** depicts the significant effect of Q (*p* = 0.0009). Data (**a**–**e**, **g**, **h**) represent two experiments with at least three separate wells per group. All error bars denote mean ± SEM

blood, we directly compared metabolite levels. Indeed, our analysis revealed many metabolites that were relatively enriched in serum or peritoneum (Supplementary Fig. 5; Supplementary Data 2). IPA analysis confirmed multiple pathways were distinct including fatty and amino acids, which were elevated in serum and peritoneum respectively (Supplementary Fig. 5). Among the most enriched amino acids in the peritoneum (Fig. 2b), we identified the neurotransmitter N-acetyl aspartate (NAA); a substrate for the pTRMØ-specific enzyme aspartoacylase (*Aspa* gene)[12]. Moreover, glutamate, was greatly enriched in the peritoneum (Fig. 2b), although glutamine, the most abundant amino acid in serum, was comparable. Glutamate, can fuel glutaminolysis for anaplerotic replenishment of the TCA cycle (Fig. 1b), suggesting importance in control of pTRMØ OXPHOS. Glutamate, like glutamine, effectively fuelled maximal OCR in pTRMØ and synergised with glutamine to enhance mitochondrial function (Fig. 2c). Collectively, these data show that peritoneal fuels can support the substantial mitochondrial function in pTRMØ. Glutamate, together with other amino acids such as glutamine, would enable maximum mitochondrial function in situ. Further, glutamine availability only limited decoupled OCR, demonstrating that pTRMØ become more dependent on glutaminolysis during stress.

**PTRMØ utilise glutaminolysis to maintain respiratory burst**. We investigated whether peritoneal-supplied, glutaminolysis-fuelled metabolism was important for a primary function of

TRMØ—respiratory burst. We monitor respiratory burst to zymosan particles using OCR. These particles, but not soluble-zymosan components or LPS, can initiate a protracted 4–6 h respiratory burst in pTRMØ (Fig. 3a, b) demonstrating that particle uptake may be necessary. Other ligands, including uric acid crystals, N-Formylmethionyl-leucyl-phenylalanine (fMLF), Pam3CSK4, R848, apoptotic thymocytes and *Saccharomyces cerevisiae* (*S. cerevisiae*) also trigger OCR increases (Supplementary Fig. 6a), although at lower magnitudes than zymosan. Therefore, we used zymosan-elicited OCR as tool to examine fuelling requirements for pTRMØ respiratory burst. Absence of glutamine or glucose did not affect zymosan phagocytosis in pTRMØ (Fig. 3c). However, zymosan-elicited OCR required glutamine and glucose to reach maximum intensity (Fig. 3d). The lower magnitude OCR increases seen with other ligands was also dependent on glutamine (Supplementary Fig. 6b). Glucose-dependent increase in ECAR was also recorded in response to zymosan (Fig. 3d). Inhibition of glycolysis and the pentose phosphate pathway (PPP) using 2-deoxyglucose (2-DG) and dehydroepiandrosterone (DHEA) confirmed the requirement for glucose metabolism (Fig. 3e), supporting that these pathways maintain the supply of ATP and/or NADPH[25]. Long-chain fatty acids were not a major requirement, and glutaminase inhibitors confirmed glutamine's effect was attributable to catabolism (Fig. 3f, g). Respiratory burst was also enhanced by glutamate, further supporting a requirement for anaplerosis (Fig. 3h). However, glutamate alone was unable to fully reconstitute the

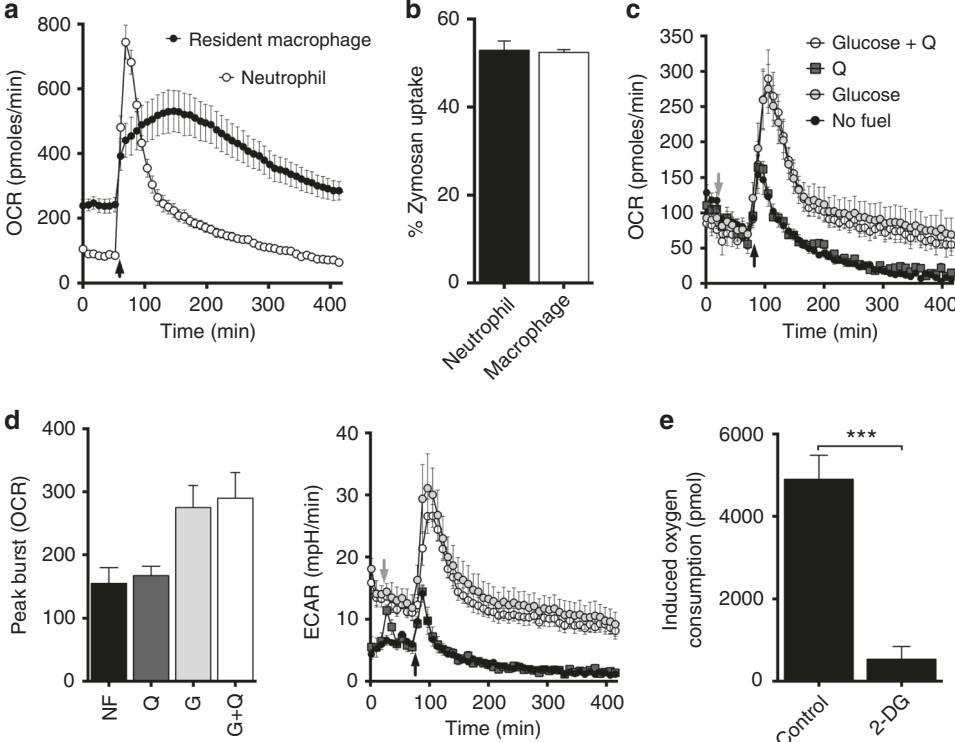

**Fig. 4** Glutamine does not support neutrophil respiratory burst. **a** Line plot of OCR vs time in peritoneal tissue-resident macrophages and bone marrow neutrophils. Arrow indicates the addition of zymosan (50 µg/ml). Data are from two separate experiments and represent at least three independent observations. **b** Bar chart showing the quantification of pHrodo zymosan (50 µg/ml) uptake. Data show three separate samples per group, represents two independent experiments and was not significant by Student's $t$-test. **c** Line plot showing OCR vs time in neutrophils in the absence of glutamine (Q) and presence or absence of glucose (25 mM) for 90 min before addition of Q (2 mM) or control media (grey arrow). Zymosan = black arrow. **d** Peak burst quantification of plots from **c** and ECAR plots from the data in **c**. Data **c**, **d** represent at two independent experiments with three separate wells per group. Peak burst data were analysed by one-way ANOVA ($p = 0.0262$), Tukey's post-tests for the addition of glutamine were not significant. **e** Total oxygen consumptions were quantified by area under the curve of OCR vs time as seen in **c**, in presence of 2-deoxyglucose (2-DG,100 mM) or vehicle control. Data ($n =$ at least 3 separate wells per group) represent two independent experiments and were analysed by Student's $t$-test. All error bars denote mean ± SEM

glutamine deprived respiratory burst, indicating different import mechanisms or efficiencies.

Collectively, these data demonstrate that respiratory burst in pTRMØ can be supported by locally enriched peritoneal glutamate and complemented by glutamine. Accordingly, the respiratory burst of human monocytes was also glutamine-dependent (Fig. 3i), indicating that macrophage metabolic programmes are not species-specific.

**Neutrophil respiratory burst is glutamine-independent.** The glutamine requirement for oxidative burst in pTRMØ was surprising, considering the glucose dependency of neutrophils[25], another professional phagocyte. Respiratory burst of neutrophils was more acute than pTRMØ (Fig. 4a), despite similar levels of zymosan phagocytosis (Fig. 4b), indicating that neutrophils initiate, but do not sustain respiratory burst. Neutrophils, unlike pTRMØ, do not require glutamine to reach their peak burst, but were wholly dependent on glucose (Fig. 4c, d) through glycolysis (Fig. 4e). An increase in glucose-dependent ECAR was also recorded (Fig. 4d). Collectively, these data demonstrate that in contrast to neutrophils, pTRMØ are pre-programmed to employ additional peritoneal fuels to augment respiratory burst with glutaminolysis.

**PTRMØ respiratory burst is enhanced by CII metabolism.** Respiratory burst was originally appreciated as a rapid increase in oxygen consumption in neutrophils in response to pathogens, but was redefined as oxidase-dependent free radical generation[26–28].

Unlike neutrophils[29], pTRMØ have substantial basal OXPHOS (Fig. 5a) that could contribute to detected OCRs. Therefore, a mitochondrial stress test was performed during zymosan-induced pTRMØ respiratory burst to precisely dissect OCR and evaluate potential mitochondrial components. Unexpectedly, OCR during zymosan-induced respiratory burst was primarily attributable to increases in basal mitochondrial parameters (Fig. 5b) rather than additional oxidase activity. However, maximal mitochondrial capacity was unchanged (Fig. 5c), demonstrating that pTRMØ have the same mitochondrial capacity regardless of stimulation, but instead engage reserve mitochondrial capacity in response to zymosan challenge. Accordingly, a NOX inhibitor had little effect on zymosan-induced pTRMØ oxygen consumption, as opposed to neutrophil which was ablated (Fig. 5d, e).

Subsequently, we dissected this unique mechanism using specific inhibitors (Fig. 5f) and utilised neutrophils, as a control to assess effects of mitochondrial inhibition on traditional NADPH oxidase-driven respiratory burst. As expected, complex I (CI) inhibition had no impact on neutrophil respiratory burst (Fig. 5g). Conversely, pTRMØ respiratory burst was highly dependent on respiratory-chain complexes I, II and III (Fig. 5h). Inhibition of ATP synthase (linker of the respiratory-chain to OXPHOS), had a less dramatic impact (Fig. 5h), suggesting the respiratory-chain is decoupled during zymosan phagocytosis, which is supported by the obeservation of increased proton leak (Fig. 5b). Unlike neutrophils, pTRMØ have substantial basal-OCR, suggesting inhibition of vital OXPHOS components may affect phagocytosis itself. Indeed, reductions in pTRMØ

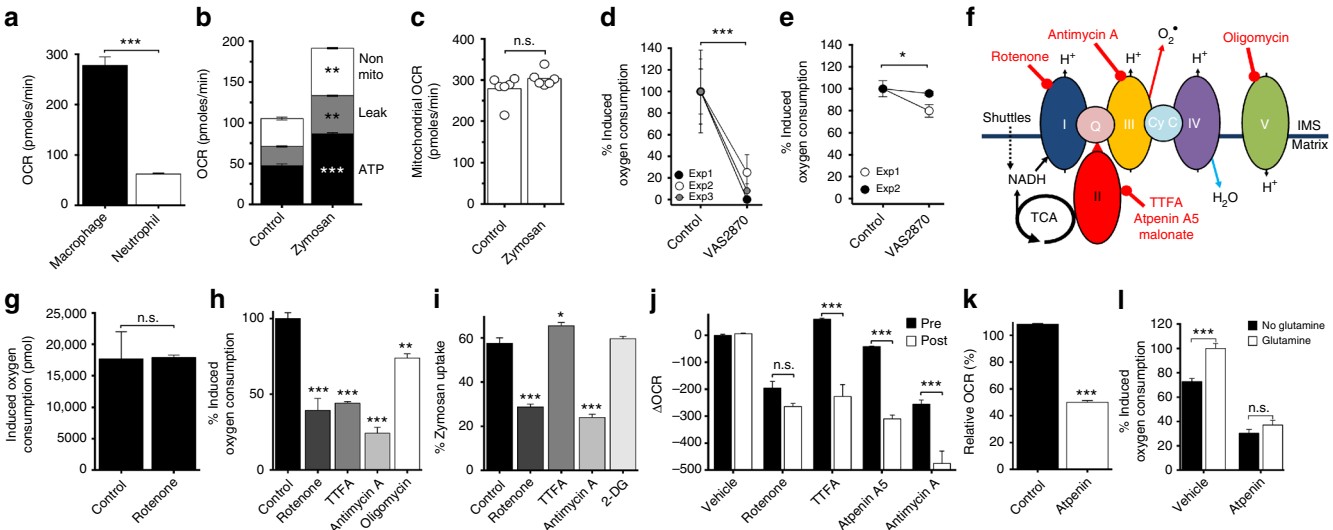

**Fig. 5** Peritoneal tissue-resident macrophages enhance mitochondrial function during respiratory burst through a glutaminolysis enhancement of complex-II. **a** Peritoneal tissue-resident macrophage (pTRMØ) and neutrophil basal-OCR. Data show at least 10 wells per group and represents three observations. **b** Changes in mitochondrial parameters after zymosan (50 μg/ml) to pTRMØ. Data ($n = 6$ separate wells per group) represent three experiments analysed by paired two-way ANOVA (interaction $p < 0.0001$) with Sidak's post-tests. **c** The maximal mitochondrial capacity from **b**. **d** Graph showing the total oxygen consumption of neutrophils after zymosan, ±VAS-2870 (10 μM). Data (9 wells per group from three experiments) were analysed by two-way ANOVA (interaction $p = 0.8567$), *** the significant effect of VAS-2870 ($p = 0.0005$). **e** Oxygen consumption of pTRMØ as in **d**. Data (6 wells per group from two experiments) were analysed by two-way ANOVA (interaction $p = 0.138$), * the significant effect of VAS-2870 ($p = 0.0331$). **f** A simplified diagram of the electron transport chain (ETC). Inhibitors are shown in red. **g** The effect of rotenone (200 nM) on total oxygen consumption of neutrophils after zymosan. Data show three wells per group and represents three experiments. **h** The total oxygen consumption of pTRMØ after addition zymosan in the presence of the ETC inhibitors: oligomycin (1.26 μM), rotenone (200 nM), thenoyltrifluoroacetone (TTFA, 2 mM), antimycin A (1 μM). Data show three wells per group, represents two experiments and were analysed by one-way ANOVA ($p < 0.001$) with Tukey's post-tests vs control. **i** The effect of metabolic drugs **f** and 2-deoxyglucose (2-DG,100 mM) on the uptake of pHrodo zymosan(50 μg/ml,1 h). Data were analysed by one-way ANOVA ($p < 0.001$) with Tukey's post-tests vs control. **j** Graph showing change in OCR after addition of indicated drugs **f** and atpenin A5 (1 μM). Post treatment (Post) is 90 min after zymosan, pre-treatment occurs before zymosan (Pre). Data show three wells per group, represents at least two experiments and were analysed by two-way ANOVA (interaction $p < 0.0001$) with Sidak's post-tests. **k** The effect of a 5 h atpenin treatment on OCR in pTRMØ. **l** Graph showing total oxygen consumption of zymosan-induced pTRMØ after atpenin A5 ±glutamine. Data (6 separate wells per group) is a pool of two experiments and was analysed by two-way ANOVA (Interaction $p = 0.0079$) with Sidak's post-tests. All error bars denote mean ± SEM, except **c** which shows the median, data in **a**, **c**, **g**, **k** were analysed by Student's $t$-test

phagocytosis were observed after CI or complex III (CIII) inhibition, but not CII (Fig. 5i). To control for this, electron transport complex inhibitors were used after zymosan uptake. The effect of these drugs on baseline OCR was compared to those recorded 90 min post-zymosan (Fig. 5j). CI inhibition had similar effects pre-and post-zymosan, indicating that CI function is necessary both before and during respiratory burst. Remarkably, CII inhibition did not substantially affect basal mitochondrial OCR, but drastically reduced the zymosan-induced OCR, demonstrating that CII engagement is enhanced during respiratory burst. Inhibition of CI or CIII had similar effects on basal-OCR. However, CIII inhibition during respiratory burst had a much greater impact, nearly eliminating both basal and zymosan-induced mitochondrial OCR. Together, these data strongly support a model wherein CI and CIII support basal pTRMØ OCR with little input from CII; however, during respiratory burst, CII engagement increases, supplying additional electrons to enhance OCR via CIII. This could affect ATP production (OXPHOS) or result in CIII-derived reactive oxygen species (ROS)[30–32]. It should be noted that, despite minimal short-term effects of CII inhibition on OCR (<1 h), long-term inhibition (5 h) did substantially suppress OCR (Fig. 5k); showing that CII is required for long-term basal metabolism, but it does not directly contribute to measured basal-OCR.

The increased utilisation of CII and the requirement for glutaminolysis are often linked. The product of glutaminolysis, α-ketoglutarate, results in TCA-supplied succinate, and glutamate can also pass through the GABA shunt to fuel CII (succinate

dehydrogenase)[33]. Indeed, we found that glutaminolysis enhancement of zymosan-induced OCR is lost after CII inhibition (Fig. 5i), demonstrating that glutamine facilitates increased pTRMØ CII activity.

Together these data demonstrate that pTRMØ are pre-programmed to exploit tissue-niche fuels to deliver a mitochondrially-driven respiratory burst through enhanced utilisation of CII.

**PTRMØ metabolic changes are independent from TLR and NO.** Recently, Garaude et al.[34] suggested that BMDM alter their mitochondrial supercomplex structures to favour CII activity after sensing bacterial RNA through *Ticam1*- and *Myd88*-dependent pathways. In contrast, decoupling the mitochondrial oxidative chain, part of the mitochondrial stress test, is sufficient for CII-dependent maximum pTRMØ mitochondrial capacity (Fig. 6a) and this dependency was unchanged after zymosan addition (Fig. 5c). Moreover, we recorded comparable levels of CII function before and after zymosan in pTRMØ (Fig. 6b). These data show, that unlike BMDM, which enhance their maximal mitochondrial capacity after activation via super-complex changes[34], zymosan-stimulated pTRMØ do not.

Although, we considered it unlikely that zymosan contains intact RNA, it does contain ligands for Toll receptors, which could stimulate similar pathways in pTRMØ. Thus, we tested whether increased CII-dependent OCR might be independent from Toll-like pathways by comparing metabolic changes after

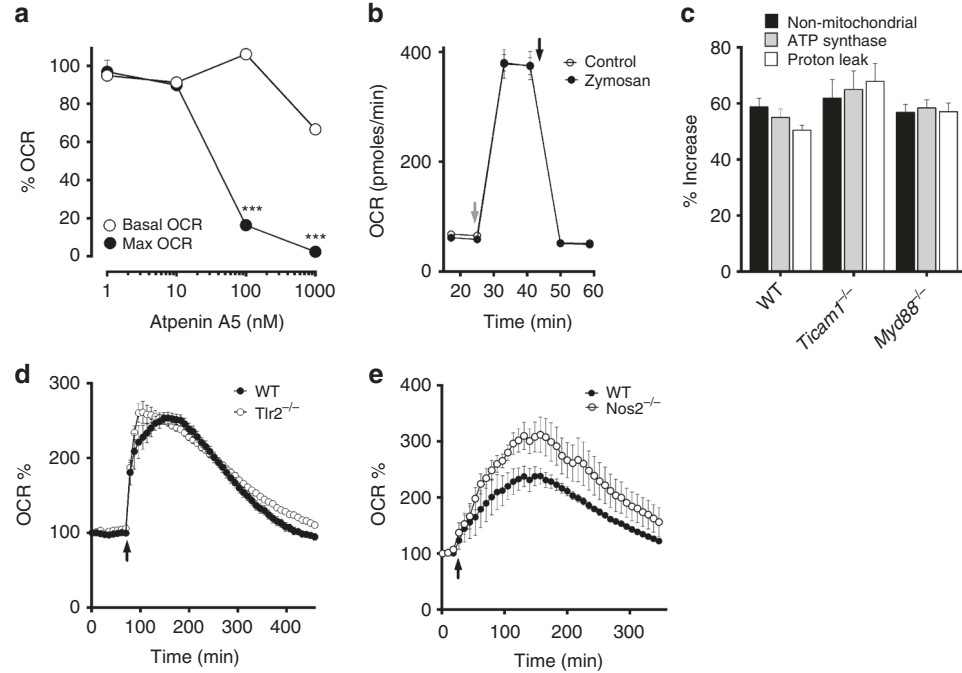

**Fig. 6** The metabolic switch in peritoneal tissue-resident macrophages is independent of nitric oxide and TLR signalling. **a** Line graph showing relative changes in oxygen consumption rate (OCR) vs atpenin A5 dose in peritoneal tissue-resident macrophages (pTRMØ). Data are from a single experiment ($n = 5$–6 separate wells per group), but the difference between basal and maximal OCR changes represents at least 3 experiments. **b** Line graph showing OCR vs time in pTRMØ. Cells were treated with or without zymosan (50 µg/ml, 2 h) before metabolic flux analysis; cell membranes were permeablised and rotenone (200 nM) added, succinate (20 mM) was added at the grey arrow, and atpenin A5 (1 µM) was added at the black arrow. Data represent at least two independent experiments and shows 6 separate wells per group. Data **a**, **b** were analysed by paired two-way ANOVA with Sidak's post-tests. **c** Graph showing the increase in mitochondrial parameters from WT, MyD88$^{-/-}$ and Ticam1$^{-/-}$ pTRMØ calculated by comparing mock and zymosan treated samples as seen in Fig. 5b. Data show $n = 6$ separate wells per group from one experiment. **d** Graph showing OCR vs time, zymosan was added to wild-type (WT) and Tlr2$^{-/-}$ pTRMØ at the arrow indicated. Data show $n = 5$ separate wells per group from one experiment. **e** Graph showing OCR vs time, zymosan was added to wild-type (WT) and Nos2$^{-/-}$ pTRMØ at the arrow indicated. Data in the left panel show $n = 5$ separate wells per group pooled from two experiments. All error bars denote mean ± SEM

zymosan in Tlr2$^{-/-}$, MyD88$^{-/-}$ and Ticam1$^{-/-}$ pTRMØ. Zymosan-induced pTRMØ metabolic changes were unaffected in these mice (Fig. 6c, d), showing that CII enhancement can occur independently of toll-like receptor/ bacterial RNA-sensing in these cells. Nitric oxide (NO) synthase (NOS) also consumes oxygen, produces free radicals, and has been shown to affect mitochondrial function[35]. However, respiratory burst was not perturbed in Nos2$^{-/-}$ pTRMØ (Fig. 6e). Taken together, these data demonstrate that although BMDM may require TLR-mediated RNA sensing for mitochondrial changes in response to bacteria, pTRMØ are poised to engage enhanced CII-mediated metabolism independently of TLR signalling.

**PTRMØ metabolic changes require protein kinase C.** Garaude et al.[34] reported that the metabolic switch to CII was entirely dependent on NOX2 in BMDM, conversely, we found that NOX2 inhibition had relatively minor effects on pTRMØ zymosan-induced oxygen consumption (Fig. 5e). However, pharmacological inhibition does not always completely eliminate activity, therefore we studied the burst in pTRMØ from mice lacking nuclear cytosolic factor-1 (Ncf1$^{-/-}$), a vital NOX2 component. These cells showed no impairment in zymosan phagocytosis (Fig. 7a). However, respiratory burst was reduced, which is remarkable considering enhanced OCR was dependent on mitochondrial function, and that inhibition had little effect (Fig. 5e). Despite these reductions, we observed a similar pattern of increased OCR in response to zymosan, albeit at lower magnitude (Fig. 7b). This suggested some NOX2 activity is required for amplification of mitochondrial changes during respiratory

burst, but is not required for engagement of CII. Indeed, enhanced oxygen consumption during the Ncf1$^{-/-}$ pTRMØ respiratory burst was dependent on CII (Fig. 7c, d).

Enhancement of CII in zymosan-induced pTRMØ respiratory burst is independent of TLR signalling, therefore we investigated other mechanisms. protein kinase C (PKC) activation occurs downstream of dectin-1/Syk signalling, which are activated by zymosan in MØ[36, 37]. However, Dectin-1 is required for zymosan recognition in phagocytes[36], so it may be required for engulfment as well as downstream signalling. To assess the role of PKC directly, we used phorbol–myristate–acetate (PMA). PMA does not interfere with dectin-1/Syk-mediated phagocytosis, but directly activates PKC. PMA treatment results in a short burst in OCR, the amplitude of which was dependent on NOX2 activity (Fig. 7b). However, PMA also enhanced mitochondrial function in Ncf1$^{-/-}$ pTRMØ (Fig. 7b), which was dependent on mitochondrial electron transport complexes (Fig. 7e), and elevated CII activity (Fig. 7f). We confirmed the requirement for PKC, through equivalent inhibition of zymosan or PMA-induced respiratory burst with a PKC inhibitor (Fig. 7f). The lower magnitude respiratory bursts seen with the addition of other ligands was, in the majority of cases, PKC-dependent (Supplementary Fig. 6c). Collectively, these data show that PKC activation can increase mitochondrial engagement of CII in pTRMØ independently of NOX2, however residual NOX2 activity may facilitate enhancement of these mitochondrial changes during phagocytosis.

**The pTRMØ metabolic switch supports the production of ROS.** CII-dependent OCR increases during respiratory burst

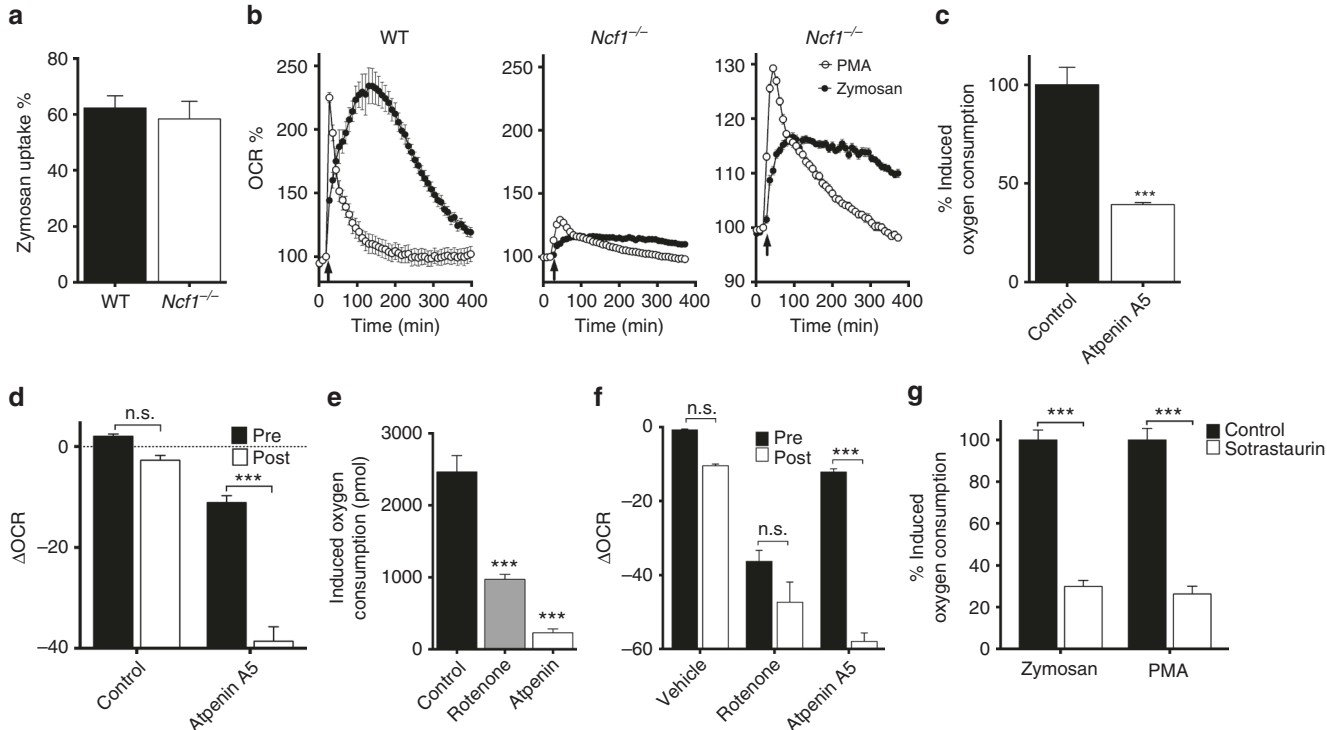

**Fig. 7** The metabolic enhancement in peritoneal tissue-resident macrophages occurs independently of NOX2, but can be initiated with activation of protein kinase C. **a** Peritoneal tissue-resident macrophage (pTRMØ) uptake of pHrodo zymosan (50 µg/ml) from $Ncf1^{-/-}$ and WT mice. Data show 3 separate wells per group and was not significantly different by Student's $t$-test. **b** OCR changes in WT and $Ncf1^{-/-}$ pTRMØ after addition of zymosan or phorbol–myristate–acetate (PMA, 1 µM) at the indicated arrow. The far-right line graph shows an expanded view of $Ncf1^{-/-}$ data from the middle plot. Data from the WT show $n = 3$ separate wells per group from one experiment, whereas the $Ncf1^{-/-}$ shows $n = 5$ per group from two separate experiments. Both are representative of at least three independent experiments. **c** The relative total oxygen consumption after addition of atpenin A5 or vehicle control in $Ncf1^{-/-}$ pTRMØ. Data ($n = 5$ separate wells per group) represent two independent experiments and were analysed by Student's $t$-test. **d** Graph showing change in OCR after addition of atpenin A5 or vehicle control. Post treatment (Post) is applied 90 min after zymosan addition, pre-treatment occurs before, in the absence of zymosan (Pre). Data ($n = 10$–12 separate wells per group) were pooled from two independent experiments and were analysed by two-way ANOVA (interaction $p < 0.0001$) with Sidak's post-tests. **e** Quantification of the total oxygen consumption after addition of PMA in the presence of the indicated inhibitors (rotenone = 200 nM). Data ($n = 3$–6 separate wells per group) represent two independent experiments and were analysed by one-way ANOVA ($p < 0.0001$) with Tukey's post-tests. **f** Graph showing change in OCR after addition of the indicated drugs. Post treatment (Post) is applied 40 min after PMA addition, pre-treatment occurs before, in the absence of PMA (Pre). Data ($n = 5$–6 separate wells per group) represent two independent experiments. Data were analysed by two-way ANOVA (interaction $p < 0.0001$) with indicated Sidak's post-tests. **g** The protein kinase C inhibitor sotrastaurin (5 µM) or control were added to pTRMØ 30 min before addition of respiratory burst stimulants PMA or zymosan, data are quantified as relative total oxygen consumed post burst stimulants. Data ($n = 5$ separate wells per group) represent two independent experiments, analysed by two-way ANOVA (interaction $p = 0.6854$, sotrastaurin $p < 0.0001$) with indicated Sidak's post-tests. All error bars denote mean ± SEM

could have consequences for cell functionality via free radical production, ATP synthesis and/or NADPH recycling. ATP is required for lysosome acidification[38] and organelle movement, whereas NADPH is utilised for NOX2 activity and maintains cellular redox states. Therefore, we assessed ATP and NADPH after fuel deprivation in pTRMØ, conditions demonstrated to alter respiratory burst after zymosan. Although ATP was drastically reduced after treatment with zymosan, absence of glucose or glutamine did not greatly affect ATP levels (Fig. 8a). NADPH was reduced after zymosan regardless of the availability of glucose or glutamine (Fig. 8a). This negligible effect of fuels on ATP and NADPH levels is distinct from patterns we observed in OCR (Fig. 3d), and contrasted with neutrophils which exhibited an absolute requirement for glucose in the maintenance of ATP levels, NADPH levels (Fig. 8b) and zymosan-induced OCR (Fig. 4c, d). The insignificant effects of fuel starvation on ATP and NADPH levels in pTRMØ demonstrates that these cells are metabolically plastic. Indeed, inhibition of metabolic pathways revealed that pTRMØ were not dependent on any singular pathway for maintenance of NADPH. However, we found

substantial dependence on glycolysis rather that OXPHOS for ATP production (Fig. 8c). Taken together, these data support the conclusion that pTRMØ have robust metabolic plasticity, which compensates for the lack of fuels, whereas neutrophils are dependent on glucose.

An alternative mechanism for the involvement of mitochondria in respiratory burst is direct contribution to respiratory burst via mitochondrial superoxide and hydrogen peroxide. Zymosan non-specifically binds many mitochondrial targeted dyes (Fig. 9c, d). Therefore, we used PMA to assess mitochondrial capacity for producing superoxide. To limit NOX-derived ROS contamination, we used $Ncf1^{-/-}$ pTRMØ. PMA treatment increased mitochondrial superoxide in $Ncf1^{-/-}$ pTRMØ (Fig. 8d). Using a hydrogen peroxide probe, we tested whether localised superoxide production in the mitochondria is indicative of general production of free radicals. Addition of zymosan to pTRMØ resulted in a hydrogen peroxide burst, which correlated with the respiratory burst measured by OCR. The hydrogen peroxide burst was sensitive to glutamine-deprivation or electron transport inhibition (Fig. 8e). These data demonstrate that complex-II dependent

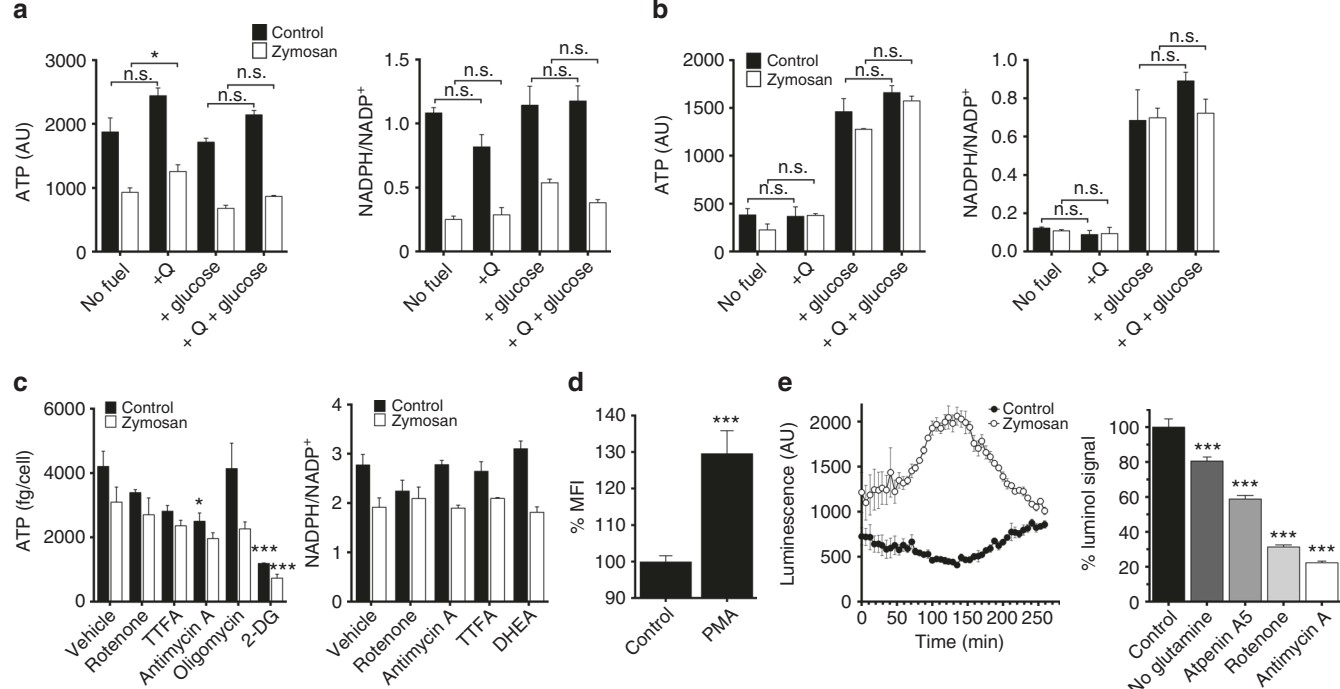

**Fig. 8** The electron transport chain is required for nominal reactive oxygen species production in peritoneal tissue-resident macrophages. **a** Bar graphs showing ATP and NADPH assays of peritoneal tissue-resident macrophages (pTRMØ). Cells were starved of glutamine (2 mM) and/or glucose (25 mM) 1 h before addition of zymosan (50 µg/ml) or control, extracts were taken after 2 h. Data were analysed by two-way ANOVA. For ATP, interaction $p = 0.16$, effects of zymosan or fuel $p < 0.0001$. For NADPH ratio, interaction $p = 0.041$. **b** Bar graphs showing ATP and NADPH assays of neutrophils. Cells were treated as in **a**, data were analysed by two-way ANOVA with Sidak's post-tests for glutamine addition. For ATP, interaction $p = 0.21$, zymosan $p = 0.15$, fuel $p < 0.0001$. For NADPH ratio, interaction $p = 0.85$, zymosan $p = 0.56$, fuel $p < 0.0001$. **c** Bar graphs showing ATP and NADPH assays of pTRMØ. Cells were treated with inhibitors 30 min before addition of zymosan as in **a**. Oligomycin (1.26 µM), rotenone (200 nM), thenoyltrifluoroacetone (TTFA, 2 mM), antimycin A (1 µM), dehydroepiandrosterone (DHEA, 100 µM), 2-deoxyglucose (2-DG, 100 mM). Data were analysed by two-way ANOVA with Sidak's post-tests vs control. For ATP, interaction $p = 0.47$, zymosan or drug were $p < 0.0001$. For NADPH ratio, interaction $p = 0.036$. Data **a**–**c** represent at least two independent experiments and show 3–4 separate wells per group. **d** Bar chart showing the median fluorescent intensity (MFI) of mitoSOX dye in $Ncf1^{-/-}$ pTRMØ treated with phorbol–myristate–acetate (PMA, 1 µM) or control. Data ($n = 4$ separate wells per group) represent two independent experiments and were analysed by Student's $t$-test. **e** Line graph (left) showing luminol luminescence against time in pTRMØ treated with zymosan 1 min before time 0. Bar chart (right) showing quantification of the area under the curve luminescence from the line graph in the presence of the indicated inhibitors (as in **b** + atpenin A5 (1 µM)). Data ($n = 4$–6 separate wells per group) were combined from two separate experiments, each treatment result is representative of at least two independent experiments. Data were analysed by one-way ANOVA with Tukey's post-tests vs the matched control. All error bars denote mean ± SEM

OCR correlates with ROS production, supporting that glutaminolysis supplies the electron transport chain to fuel pTRMØ mitochondrial ROS production during the response to zymosan.

**PTRMØ CII metabolism fine-tunes cytokine production.** MØ-derived cytokines are secreted in response to microbes or tissue damage and control inflammation[1]. TRMØ produce inflammatory mediators in response to *S. cerevisiae*, the microorganism from which zymosan is derived. Using two inhibitors, we measured specific effects of CII inhibition on *S. cerevisiae*-induced pTRMØ cytokine production. Interestingly, CII inhibition consistently reduced production of interleukin (IL)-10 and tumour necrosis factor (TNF), whereas IL-1β (mRNA and mature protein) and the neutrophil chemoattractant CXCL-1 (KC) were unaffected or inconsistently changed (Fig. 9a, b). Surprisingly, a different CII inhibitor, dimethylmalonate, had no effect on *Il1b* mRNA, but blocked ATP-induced secretion of IL-1β (Fig. 9a, b). Collectively, this shows that CII activity is required to fine-tune macrophage functional responses, which predictably will impact inflammatory resolution.

**PTRMØ mitochondria are recruited to the cytotoxic phagosome.** We next investigated whether pTRMØ mitochondria associate with the phagolysosome, which would allow mitochondria to directly affect phagolysosome content, as has been reported in BMDM and monocytes[39, 40]. The pTRMØ mitochondrial network is vast and tubular (Fig. 9c; Supplementary Movies 3–4) and ingestion of zymosan did not alter mitochondrial size significantly (Fig. 9f). However, after phagocytosis, zymosan is often situated near mitochondrial networks (Fig. 9c), which may represent mitochondrial recruitment to the phagolysosome[39]. Indeed, in some cases, as reported[40], three-dimensional cupping of phagolysosomes with mitochondria was observed (Fig. 9d). Linear regression analysis confirmed enrichment of mitochondria toward the phagolysosome (Fig. 9e, $R^2 = 0.7162$, non-zero $p < 0.0001$) making them well-situated to control the outcome of pTRMØ phagolysosome function.

We examined whether mitochondrial function was important for the killing of phagolysosome contents. Unexpectedly, addition of the CII inhibitor did not significantly reduce the killing of *S. cerevisiae* by pTRMØ (Fig. 9g), as has been reported in the BMDM killing of bacteria[34]. This could be attributed to residual mitochondrial function after CII inhibition, which is capable of producing sufficient ROS to kill *S. cerevisiae,* while we find

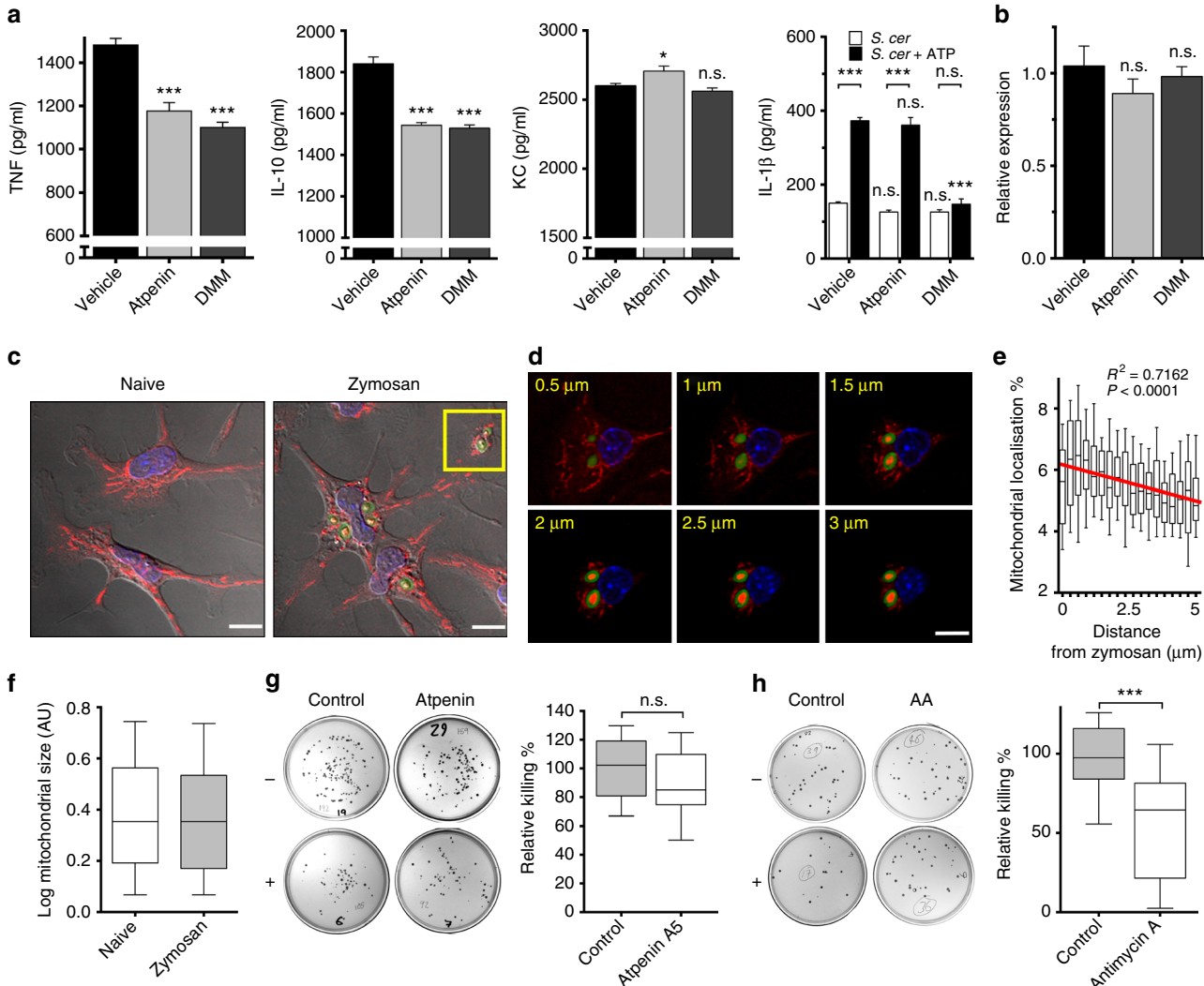

**Fig. 9** The mitochondrial electron transport chain is required for antimicrobial function in peritoneal tissue-resident macrophages. **a** Quantification of cytokines from peritoneal tissue-resident macrophage (pTRMØ) supernatant 24 h after *Saccharomyces cerevisiae* (*S. cerevisiae*) (50 µg/ml) ±dimethylmalonate (DMM,10 mM) or ±atpenin A5 (1 µM). The far-right panel shows IL-1β after 1 h post-stimulation with adenosine triphosphate (ATP, 3 mM), 23 h after *S. cerevisiae*. Data (*n* = 4 separate wells per group) represent two independent experiments. For TNF, IL-10 and KC, data were analysed by one-way ANOVA with Dunnett's post-tests; for IL-1β, data were analysed by two-way ANOVA with Sidak's post-tests vs vehicle control or vs no ATP as indicated (interaction *p* < 0.0001, ATP *p* < 0.0001, complex-II *p* < 0.0001). Error bars = mean ± SEM. **b** Relative expression of *Il1b* RNA from samples in **a**. **c** Representative photographs of pTRMØ. In **c**, **d**, zymosan-AlexaFluor488 (5 µg/ml) are green, mitochondria labelled with Mitotracker Red CMX-Ros (25 nM) are red, whereas the nuclei are labelled blue with 6-diamidino-2-Phenylindole (DAPI,125 ng/ml). The yellow box denotes mitochondria surrounding zymosan. Zymosan cores can be seen with non-specific Mitotracker Red fluorescence. **d** A characteristic *z*-stack showing mitochondria surrounding the phagolysosome. The yellow numbers denote distance from coverslip. **e** Quantification of the number of Mitotracker Red positive pixels expressed as percentage of total (termed mitochondrial localisation %) at different radial distances from zymosan. Data shown are 53 zymosan particles from five images. Data were analysed by a one-way ANOVA (*p* < 0.0001) with a linear regression post-test shown. **f** Box and whisker plots showing quantification of size for >800 mitochondrial units per group taken from five images of pTRMØ in the presence or absence of zymosan-AlexaFluor488. **g** Representative pictures (left) of microbial plates with *S. cerevisiae* colonies. Data shown are from *S.cerevisiae* cultured ± pTRMØ with atpenin A5 (1 µM) or control for 4½ h. The graph on the right shows quantification of the average microbial killing. Data are from three experiments (*n* = 13 per group) and were analysed by two-way ANOVA (interaction + experimental repeat *p* = 0.26, atpenin A5 *p* = 0.10). **h** Representative pictures (left) and quantification of data (right) as in **e**, but with antimycin A (1 µM) or control. Data are from four experiments (*n* = 18 per group) and were analysed by two-way ANOVA (interaction + experimental repeat *p* = 0.10, antimycin A *p* < 0.0001). Whiskers **e–h** show 10–90% of the range. White scale bars are 10 µm

that CIII is required for most pTRMØ ROS. Thus, in contrast to CII, CIII inhibition had a marked impact, even when added post *S. cerevisiae* to permit unadulterated uptake (Fig. 9h). This strongly suggested that reduced free radical production in the absence of CII activity is sufficient for antimicrobial function, whereas CIII inhibition may reduce mitochondrial function below the threshold required for effective antimicrobial activity. Collectively, these data suggest that pTRMØ mitochondria move toward phagolysosomes

where mitochondrial function contributes to the killing of ingested microbes.

## Discussion
Relatively little is known about TRMØ metabolism. Past research revealed that human leucocyte phagocytosis correlates with glutamine availability in the blood[41], and that glutamine was

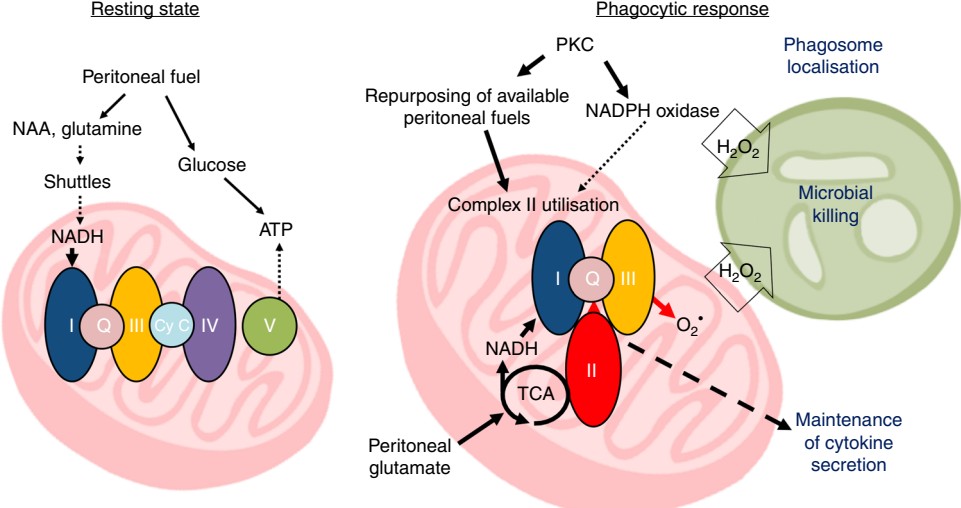

**Fig. 10** Peritoneal tissue-resident macrophages are metabolically poised to engage microbes using tissue-niche fuels Model showing the proposed metabolic phenotypes of resting- and phagocytic-peritoneal tissue-resident macrophages and how this impacts function. NAA, N-acetyl aspartate; PKC, Protein kinase C; Shuttles, malate-aspartate or citrate-malate mitochondrial shuttle systems

important for interleukin production[42, 43]. Newsholme, Gordon and colleagues demonstrated that pTRMØ have an innate ability to use glutamine, but do not fully utilise this capacity during homoeostasis[24, 44]. Therefore, we hypothesised that pTRMØ glutamine metabolism could be maximised during stress, and might be limited by glutamine availability in the tissue environment.

In vitro-derived BMDM have been routinely used to study MØ functions, however TRMØ have different origins and are at the front-line of tissue defence. We hypothesised that examining these cells in context with their niche environment might reveal unique characteristics and/or mechanisms[2]. On the basis of this approach, we propose a model (Fig. 10) where peritoneal-enriched fuels support metabolic repurposing of pTRMØ during stress. PTRMØ have considerable mitochondrial capacity fuelled by peritoneal glutamate/ glutamine, which supports CII-dependent repurposing of mitochondrial metabolism after phagocytosis and during stress. This metabolic switch is distinctive to a mechanism described in in vitro-derived BMDM[34]; can be initiated through PKC activation, and is required for pTRMØ ROS. Furthermore, we demonstrate that pTRMØ mitochondrial function is required for nominal cytokine expression and microbial killing.

PTRMØ have increased TCA metabolites, larger mitochondrial networks and a greater OXPHOS capacity compared to in vitro-derived BMDM. RNA-seq analysis did not uncover a causal basis for these differences, though it revealed opposite metabolic isoform expression patterns between these populations. Differential utilisation of metabolic transcriptional programmes may underlie distinct mitochondrial phenotypes and function, although more study is needed to dissect these mechanisms. Alternatively, mitochondrial differences we describe may be the result of cell-specific reactions to non-physiological oxygen and metabolic fuels in vitro or reflect different rates of proliferation. Mitochondrial network volumes are known to flux as the cell-cycle progresses[45]. As the proliferation rate of pTRMØ is very low[3] compared to that of BMDM, we cannot rule out that our findings are a reflection of "resting" vs "proliferative" mitochondria. Despite these caveats, when taken together with our metabolic profiling, these data show that in vitro-derived MØ are phenotypically and metabolically distinct to pTRMØ, suggesting their specialised metabolic

programming is vital for cell-specific functions in unique environments in situ.

We found the peritoneal metabolome to be distinct from serum, containing elevated amino acids, especially glutamate. Glutamate supplements glutamine's ability to support pTRMØ mitochondrial function, demonstrating that glutamate production is limiting for pTRMØ metabolism in situ. The peritoneal metabolome also supports the proposed role of *Aspa* for pTRMØ survival[12], by demonstrating that, compared to serum, the peritoneum is enriched with the *Aspa* substrate NAA. Thus, we now know of two examples where the niche environment provides factors required to specifically control pTRMØ phenotype and function. Interestingly, glutamate is also a neurotransmitter. The source of peritoneal glutamate and NAA is unknown, although they may be released from neurons and/or the gut micobiome[46]. Regardless, substantial presence of glutamate and NAA in the peritoneum is consistent with the known interplay of parasympathetic nerve function, gut physiology and inflammation.

We find that zymosan-induced pTRMØ respiratory burst is glucose-dependent, fatty acid-independent and required gluta-minolysis, a metabolic pathway supported by peritoneal-enriched glutamate. Surprisingly, zymosan-induced increases in OCR were attributable to increases in mitochondrial function. This explains the glutaminolysis requirement and it demonstrates that pTRMØ respiratory burst is fundamentally different from that of neutrophils, where increases in OCR can be entirely attributed to NOX activity, which requires glucose-fuelled maintenance of NADPH[25]. In pTRMØ, respiratory burst was not dependent on opsonisation and TLR2 was not required, but other receptors such as dectin-1 may be important[36, 47], as both PMA and zymosan initiated mitochondrial-dependent respiratory bursts that were PKC-dependent. Interestingly, pTRMØ respiratory burst was accompanied by a burst in ECAR, suggesting that enhanced glycolysis may also be important; likely to supply additional ATP required for cell activation, which is consistent with the glycolytic-dependency of ATP production (Fig. 8c).

The metabolic reprograming toward increased CII metabolism in pTRMØ we describe was not dependent on live bacterial sensing, as in BMDM[34]. Rather our data suggest that pTRMØ are metabolically poised to switch toward increased CII activity in situ. A priming event, such as sensing of the microbiome[48],

may be required for this in vivo. Such signalling is absent in cultured BMDM, but could be triggered via TLR-sensing of live bacteria in vitro[34]. We demonstrate that pTRMØ enhancement of complex-II activity does not require the NOX2 component NCF-1, despite a reduced amplitude in the absence of NCF-1. Notably, we find distinct differences between biochemical suppression of NOX2 and genetic ablation; the former having little effect, whereas the latter effectively reduces burst. This seemingly paradoxical finding suggests that other cellular oxidases could specifically allow for the shift in mitochondrial function in the absence of NCF-1, whereas NOX2 may amplify burst via positive feedback, or be needed for a priming event in vivo that is not effectively inhibited pharmacologically. Such a model is supported by examples of NOX2 activating other oxidases and facilitating the generation of mitochondrial ROS[30, 49]. It is likely that glucose is required for PPP-fuelling of NOX2, as in the absence of glucose, but presence of glutamine, there is a reduced zymosan-induced OCR enhancement. Therefore, we propose that pTRMØ use glucose to activate NOX2 and maintain ATP, however the increased oxygen consumption is derived from NOX2-amplification of mitochondrial function and fuelling from glutamine, rather than NOX2 activity itself. Incidentally, we detected glutamine- and PKC-dependent increases in OCR from other ligands. However, these were of much lower magnitude than zymosan- or PMA-induced OCR. This could be attributed to a lack of NOX engagement, but could also indicate lower PKC activation or reduced uptake/binding of ligands.

Respiratory burst was not detected in BMDM, despite similar rates of zymosan uptake (Supplementary Fig. 7), perhaps due to long-term suppressive effects of exogenous M-CSF on NOX2 components in BMDM[50] or other downstream signalling mechanisms. Accordingly, cultured human monocytes also undergo a glutaminolysis-dependent respiratory burst, but only after interferon gamma pre-treatment to maintain NOX expression in vitro[51]. Thus, the respiratory burst enhancement through glutaminolysis we describe is not organism restricted, but may require NOX for optimum effectiveness.

Respiratory burst $H_2O_2$ production in pTRMØ may be supported by mitochondrial superoxide via CIII[30, 32], or the NOX-fuel NADPH via mitochondrial shuttles[52, 53]. Yet, we find that NADPH levels did not correlate with OCR changes, suggesting that mitochondrial ROS is responsible for glutaminolysis-enhanced $H_2O_2$. The near total sensitivity of zymosan-induced pTRMØ ROS to antimycin A is consistent with electron leak from CIII, rather than reported reverse electron transport from CII to CI[54]. We find that pTRMØ do not require CII to maintain their basal-OCR short-term, but increase their dependence on CII after phagocytosis. In fact, OXPHOS through CI does not require the full TCA cycle or CII, but purely requires NADH that can be supplied by mitochondrial shuttles[55]. Inhibition of OCR after long-term CII inhibition is likely attributable to a build-up of TCA metabolites, which could ultimately inhibit NADH availability and subsequently affect CI activity.

In addition to effects on respiratory burst, the switch toward increased CII activity in pTRMØ was also required for optimal cytokine secretion. Interestingly, ATP-induced IL-1β secretion was reduced after treatment with dimethylmalonate but not by atpenin A5 (CII inhibitors), whereas TNF and IL-10 were suppressed by both. Unlike atpenin A5, dimethylmalonate is a competitive inhibitor, which could affect succinate transporters and succinylation reactions. It is also known to inhibit citrate synthase[56] and all these processes have been implicated in regulation of IL-1β making its effects difficult to interpret[54]. In addition, these inhibitors are likely to alter ROS, a signal known to activate the inflammasome and drive cytokine expression[54, 57]. NO production is another possible explanation for effects on

inflammatory pathways. We have previously shown that LPS drives NO production, which directly suppresses OXPHOS in BMDM[35], reportedly by the suppression of mitochondrial complexes[58]. Therefore, inhibition of CII in BMDM could affect many factors, including NO production or its effect. However, pTRMØ stimulated with zymosan or *S. cerevisiae* do not produce appreciable NO (Supplementary Fig. 8), which is perhaps related to high arginase expression[11] known to inhibit NOS[59]. Thus, unlike BMDM, pTRMØ metabolic control in situ is likely to be largely independent of NO, which will affect the outcome of cell activation and inflammation.

Consistent with a role in respiratory burst, we found that pTRMØ recruit mitochondria toward phagolysosomes as has been shown in BMDM[39, 40, 60], the latter dependent on TLR signalling. Interestingly, mitochondrial recruitment to zymosan-phagolysosomes was maintained in $Tlr2^{-/-}$ pTRMØ (Supplementary Fig. 9), confirming their predisposition for response. Complete inhibition of mitochondrial function and subsequent ROS production using a CIII inhibitor impaired pTRMØ killing of microbes. It has been suggested that ROS in neutrophils primarily activates granule enzymes rather than directly killing microbes[61]. Given that ROS production is lower in MØ than neutrophils, it is unlikely that ROS production from MØ directly kills microbes, but rather, as with neutrophils, acts as a secondary messenger. Regardless, we find that peritoneal metabolites, which fuel pTRMØ mitochondrial function are important in the response to and the killing of microbes.

In summary, for the first time, we examine TRMØ metabolic requirements with consideration of the corresponding tissue-niche environment. We demonstrate that pTRMØ are uniquely poised to utilise tissue-niche fuels to defend against microbial challenge. Further understanding of the metabolic requirements of effector cells in their unique tissue-niches will have meaningful impacts on our approach to treating disease in a tissue-dependent manner.

## Methods

**Reagents.** All reagents were from Sigma unless otherwise stated. Specific kits and linked reagents are described below. Non-canonical reagents are listed.

Rotenone, dehydroepiandrosterone (DHEA), thenoyltrifluoroacetone (TTFA), antimycin A, 2-deoxyglucose (2-DG), oligomycin, glutamate (pH 7.4 with NaOH), zymosan particles from *Saccharomyces cerevisiae* (including pHrodo, Thermo-Fisher), Yeast from *Saccharomyces cerevisiae*, LPS (lipopolysaccharide), luminol, DTAC (Dodecyltrimethylammonium chloride), Accutase, diazo-oxo-norleucine (DON), bis-phenylacetamido-thiadiazolyl-ethyl sulphide (BPTES), atpenin A5 (Cayman chemicals), dimethylmalonate, UK-5099, etomoxir, phorbol–myristate–acetate (PMA), mitotracker red CMX-Ros (Thermo-Fisher), all-trans retinoic acid, M-CSF (Peprotech), VAS-2870, Sotrastaurin (Cayman chemicals), N-Formylmethionyl-leucyl-phenylalanine (fMLF), Pam3CSK4 (Invivogen), uric acid crystals, R848 (Invivogen), 4′,6-Diamidino-2-Phenylindole (DAPI, Thermo-Fisher), Hoescht 33342 (Thermo-Fisher), 2-mercaptoethanol.

**Mice.** Mice were on the C57BL/6 background and maintained in the Frederick National Laboratory Core Breeding Specific Pathogen Free Facility. NOX2 deficient NCF-1$^{-/-}$ ($Ncf1^{-/-}$) mice[62] were a kind gift from Dr. Steven Holland (National Institute of Allergies and Infectious Diseases, Bethesda, USA). $Nos2^{-/-}$ mice[63] were a generous gift from Dr. Victor Laubach (University of Virginia). $Myd88^{-/-}$ mice[64] were a gift from Dr. Giorgio Trinchieri, National Cancer Institute (NCI), Bethesda. $Ticam1^{-/-}$ mice were backcrossed onto the C57BL/6 background in our core facility[65]. Animal care was provided in accordance with the procedures in, "A Guide for the Care and Use of Laboratory Animals". Ethical approval for the animal experiments detailed in this manuscript was received from the Institutional Animal Care and Use Committee (Permit Number: 000386) at the National Cancer Institute-Frederick. Experiments were carried out on a mixture of male and female mice between 6 and 16 weeks of age.

**Primary cell preparation and purification.** Bone marrow leucocytes[66] and peritoneal cells[11] were isolated as previously described. Briefly, for peritoneal cells, the peritoneum of a mouse was lavaged with 5–10 ml phosphate buffered saline (PBS) using a 21-G needle. For bone marrow leucocytes, bone marrow was flushed out of trimmed leg bones using PBS and a 25-G needle. Red blood cells were lysed before

cell culture. BMDM were differentiated from these cells and maintained using 20 ng/µl recombinant mouse M-CSF as previously described[66].

For magnetic purification of primary cell populations: Cells were blocked with 4 µg/ml α-FcγIII (2.4G2, in house) in wash buffer PBS with 5 mM ethylenediaminetetraacetic acid (EDTA), 0.5% bovine serum albumin (BSA)) for 5 min on ice, before addition of 20 µg/ml α-Ly-6G-Biotin (1A8, Biolegend) to label neutrophils or 20 µg/ml α-F4/80-Biotin (BM8, Biolegend) to label pTRMØ for 30 min on ice. Cells were then magnetically sorted with streptavidin micro-beads and LS columns as per manufactures instructions (Miltenyi Biotec), with an additional wash of 3 ml complete media (Dulbecco's minimal essential media (DMEM) with no pyruvate, 10 % fetal calf serum (FCS), 0.2 units/ml penicillin, 100 µg/ml streptomycin, 2 mM L-glutamine, 25 mM glucose) (Thermo-Fisher) for pTRMØ, or Seahorse media (Seahorse Bioscience) for neutrophils to remove EDTA, before elution of the positive selection in complete media or Seahorse media. Purities were typically 90–95% for neutrophils and 85–90% purity for pTRMØ, with the major peritoneal contaminant primarily being F4/80⁺ eosinophils.

Human monocytes were obtained by gently adding 35 ml blood to 15 ml Histopaque (Ficoll) in a 50 ml centrifuge tube. The tube was centrifuged for 30 min at 600×g at room temperature. The peripheral blood mononuclear cell (PBMC) layer was removed to a centrifuge tube and re-suspended to 50 ml with cold PBS (+2 mM EDTA). This was again centrifuged for 5 min at 600×g and the supernatant removed, the cells were washed two more times with PBS (+2 mM EDTA). The Percol kit (GE Healthcare, 17-0891-01) was used to purify monocytes. Briefly, PBMCs were re-suspended in 5 ml of solution A and this solution gently layered into a 15 ml centrifuge tube containing a 54:47 mixture of solution A to solution B. The tubes were centrifuged for 30 min at 1000×g, and the monocyte band removed, re-suspended in complete media and counted. Human blood was obtained from healthy volunteers, who were recruited through the National Cancer Institute-Frederick Research Donor Program and provided written informed consent. All users of human materials were approved and appropriately trained.

**Extracellular flux analysis.** For neutrophil adherence, wells of an extracellular flux (XF) tissue culture plate were treated with Cell-Tak solution as per manufacturer's instructions (Corning), however this was found to be unnecessary for adherence. Magnetically sorted pTRMØ were seeded at 0.5–1.0 × 10⁶ (24-well) or 0.25 × 10⁶ cells per well (96-well) in complete media (+0.5 µM retinoic acid, + 20 ng/ml M-CSF) and incubated for 1.5–2 h. The media was removed and replaced with Seahorse assay media with 2 mM glutamine and 25 mM Glucose (+0.5 µM retinoic acid, +20 ng/ml M-CSF) unless stated. Magnetically sorted neutrophils (1.0 × 10⁶ cells per well, 24-well) in Seahorse assay media were also added to the Seahorse plate. The neutrophils were centrifuged to the bottom of the well at the lowest acceleration to 45×g followed by natural deceleration. Fully differentiated BMDM were removed from plates with a 10 min incubation of accutase solution (37 °C) and firm washing. Cells were centrifuged for 5 min at 350×g, the supernatant removed and the cells re-suspended in Seahorse media containing 20 ng/ml M-CSF, before plating for seahorse analysis: 100,000 cells for 96-well, 400,000 cells for 24-well. Human monocytes (above) were cultured for 48 h at 2-million cells/ml in the presence of recombinant human M-CSF (50 ng/ml) and interferon-γ (10 ng/ml). Monocytes were removed from plates like BMDM above, with the exceptions, that recombinant human M-CSF (50 ng/ml) and interferon-γ (10 ng/ml) were added to the Seahorse media and cells were plated in the same concentrations as pTRMØ. The plates containing cells were incubated for 1 h at 37 °C with no CO₂. XF analysis was performed at 37 °C with no CO₂ using the XF-24 or XF-96 ana-lyzer (Seahorse Bioscience) as per manufacturer's instructions. Port additions and times were used as indicated in the figures. Lavaged media was also used to replace media on pTRMØ, briefly, 0.5 ml of Seahorse XF base media (no glucose or glu-tamine) was used to lavage peritoneal mice, with extra mixing. This lavage media was centrifuged at 500×g for 10 min, to remove cells, the supernatant removed and added to an Amicon 3 kD centrifugal filter (Millipore). This was centrifuged at 3000×g for 60 min, to remove large proteins.

For apoptotic thymocytes: Thymuses were extracted and mechanically dissociated in complete RPMI media. Cells were re-suspended and counted in complete media. Thymocytes were seeded in 6-well plates (10 × 10⁶ per well) with dexamethasone (1 µM) for 3 h. Cell were extracted from plates and washed three times in seahorse base media before re-counting and loading into injection ports (500,000 cell per well). Cells were additionally labelled using the PE Annexin V Apoptosis Detection kit (Becton Dickinson) as per manufacturer's instructions. On average apoptosis was recorded as 20–40% exposure of phosphatidylserine with <10% dead cells.

**Flow cytometry.** For zymosan uptake: pHrodo zymosan particles (50 µg/ml) was added to cells for 1 h or mitoSOX Red (2.5 µM) was added to cells for 20 min. Cells were washed with PBS and detached from culture plates using Accutase treatment at 37 °C for 15 min, before thorough pipetting. Cells were transferred to flow cytometry tubes and analysed by a BD LSRII or BD Fortessa flow cytometer for fluorescent analysis. Doublets and debris were gates out before quantification of median fluorescent intensities with FlowJo (FlowJo, LLC).

For apoptosis detection: Cells were transferred to flow cytometry tubes and analysed by a BD LSRII flow cytometer. Doublets were gates out before evaluation

of % Annexin V⁺ 7AAD⁻ (apoptotic) and % Annexin V⁺ 7AAD⁺ (dead) with FlowJo (FlowJo, LLC) (Supplementary Fig. 10).

For purity checks: α-F4/80-Pacific blue/ Alexa Fluor 647 (BM8, Biolegend) or α-CD11b allophycocyanin-Cy7 (M1/70, Becton Dickinson) were used for pTRMØ, and α-Ly-6G (1A8, Biolegend) or CD11b allophycocyanin-Cy7 was used for bone marrow neutrophils (Supplementary Fig. 10).

**Real-time polymerase chain reaction and RNA sequencing.** For real-time polymerase chain reaction: Atpenin A5 or dimethylmalonate were added 15 min before 50 µg/ml serum-opsonized (15-min serum opsonization at 37 °C) S. cere-visiae to 600,000 pTRMØ in wells of a 48-well plate (final volume 250 µl). The plate was incubated for 24 h before RNA was extracted using the High-Pure RNA iso-lation kit (Roche, 11828665001), as per manufacturer's instructions (with the addition that 2-mercaptoethanol (1% v/v) was added to the lysis buffer). The High-Capacity cDNA Reverse Transcription kit (Thermo-Fisher, 4368813) was used as per manufacturer's guidelines to generate cDNA from the RNA. Real-time PCR was performed on an Applied Biosystems 7300 using Eagle Taq (Roche) reagent and TaqMan gene expression probes: Hprt (Mm01545399_m1), IL-1b (Mm00434228_m1) (Thermo-Fisher). Values are expressed as relative expression using the ΔΔCt method: IL-1b cycle threshold number (Ct) minus Hprt Ct = ΔCt, minus control sample ΔCt = ΔΔCt, relative expression considering the doubling per cycle of PCR = $2^{-\Delta\Delta Ct}$.

For RNA-sequencing: RNA was taken from 1×10⁶ naïve BMDM and pTRMØ using the method above. RNA quality was assessed using Agilent BioAnalyzer Nano chips and the RNA Integrity Number (RIN) algorithm. The RIN number was 8.3, 10, 10 and 10 for two pTRMØ and two BMDM samples respectively. RNA-Seq libraries were constructed from 1 µg of RNA using the Illumina TruSeq Stranded mRNA library preparation kit (Illumina, CA). Sequencing was done on Illumina HiSeq 2500 sequencers, with two lanes of paired-end 125 bp reads generated per sample. Raw files were aligned to mouse genome (mm10) using STAR (v. 2.3.0)[67]. Normalisation was performed using trimmed mean of end values (TMM)[68] after filtering out the genes that have five raw counts or less and were not shared by at least one sample. The Limma-Voom[69] paired test was used to obtain fold values.

**Luminescence assays.** Magnetically sorted pTRMØ were seeded at 0.2–0.4 × 10⁶ cells per well of a 96-well luminescence plate in complete media and incubated for 2 h at 37 °C. The media was replaced with minimal media (DMEM with no pyr-uvate, no phenol red, no glucose, no glutamine, 0.2 units/ml penicillin, 100 µg/ml streptomycin, or Seahorse media (no glutamax, no glucose, no glutamine). Neu-trophils were then added to separate wells on the plate and cells were supplemented with ±glucose (25 mM) or ±L-glutamine (2 mM), or additional treatments as indicated (including 400 µM luminol). The plate was incubated at 37 °C for 30 min, after which ±zymosan (50 µg/ml) was added. For luminol measurements, the plate was placed in a 96-well luminescence plate reader and luminescence readings were measured every 6 min for 6 h. Otherwise, cells were lysed in passive lysis buffer (Promega) or 0.1 M NaOH with 0.5% DTAC for ATP or NADPH determination, respectively. Relative ATP was determined using the ATP assay kit (Abcam/Thermo-Fisher) and NADPH/NADP ratios determined using the NADPH-Glo assay kit (Promega) as per manufacturer's instructions.

**Primary metabolite analysis.** Magnetically sorted pTRMØ or gently scraped cultured BMDM (5.0–7.5 × 10⁶) were pelleted and snap-frozen in liquid nitrogen. A trypan blue dilution assay was used to determine the average fluid quantity in the peritoneum (Supplementary Fig. 11). This was determined to be ≈100 µl. Mice were lavaged thoroughly with 300 µl of PBS to give a ≈4 × diluted lavage sample. Blood collected in serum tubes and lavage samples were centrifuged at 1200×g for 15 min to remove cellular components. Serum was also diluted in PBS to give 4 × diluted serum samples. Samples were further processed and analysed at the West Coast Metabolomic Center (University of California, Davis). Briefly, samples were re-suspended with 1 ml of extraction buffer (37.5% degassed acetonitrile, 37.5% isopropanol and 20% water) at −20 °C, centrifuged and decanted to complete dryness. Membrane lipids and triglycerides were removed with a wash in 50% acetonitrile in water. The extract was aliquoted into two equal portions and the supernatant dried again. Internal standards C08-C30 fatty acid methyl esters were added and the sample derivatized by methoxyamine hydrochloride in pyridine and subsequently by N-methyl-N-trimethylsilyltrifluoroacetamide for trimethylsilyla-tion of acidic protons. Gas chromatography-time-of-flight analysis was performed by the Agilent GC6890/LECO Pegasus III mass spectrometer. Samples were additionally normalised using the sum of peak heights for all identified metabolites.

**Cytokine production and Saccharomyces cerevisiae killing.** For cytokine analy-sis: Atpenin A5 or dimethylmalonate were added 15 min before 50 µg/ml serum-opsonized (15-min serum opsonization at 37 °C) Saccharomyces cerevisiae (S. cerevisiae) to 600,000 pTRMØ in wells of a 48-well plate. The plate was incubated for 24 h before supernatant removal. The BD cytometric bead array kit (558267) was used to quantify cytokines in supernatant, and was used as per manufacturer's instructions. For killing quantification: S. cerevisiae was added to pre-cultured pTRMØ (described above) or empty wells in a 1:2 ratio. Antimycin A was added was added 30 min after S. cerevisiae to avoid known defects in uptake. S. cerevisiae

with or without pTRMØ were mixed with Triton-X100 (1% final) by pipetting and incubated for 5 min. Dilutions in water (1:10, 1:50, 1:200 final) were plated (25 µl) onto YPD-agar 9 cm bacterial plates using L-shaped spreaders. Plates were incubated at 30 °C for 48 h before counting of colony numbers.

**Confocal microscopy.** For Fig. 9: pTRMØ were pre-cultured in complete media (750,000 per 35 mm dish, + 0.5 µM retinoic acid, + 20 ng/ml M-CSF) for 18 h on glass-bottom culture dishes (MatTek, #0). pHrodo zymosan particles (50 µg/ml) were added to cells for 1 h and/or Mitotracker Red CMX-Ros (25 nM) was added to cells for 20 min. Cells were washed twice with PBS and fixed with 4% paraformaldehyde for 20 min at room temperature. Cells were washed with PBS and 6-Diamidino-2-Phenylindole (DAPI, 125 ng/ml) added. A Zeiss UV-510 confocal microscope and 63x oil immersion objective lens was used to capture images using differential interference contrast, 405, 488 and 561 nm laser paths (pinhole size = 1 airy unit).

For supplemental Figs. 4 and 7: pTRMØ (300,000 per 24-well, +0.5 µM retinoic acid, +20 ng/ml M-CSF) or BMDM (150,000 per 24-well, +20 ng/ml M-CSF) were seeded for 18 h on glass-bottom culture dishes (CellVis, #1.5). pHrodo zymosan particles (50 µg/ml) were added to cells for 1 h and/or Mitotracker Red CMX-Ros (25 nM) was added to cells for 20 min. Cells were washed twice with PBS and fixed with 4% paraformaldehyde for 20 min at room temperature. Cells were washed with PBS and Hoescht 33342 (1 µg/ml) added before further washing. A Leica TCS SP8 confocal microscope with Yokogawa CSUW1 spinning disk and a ×100 oil immersion objective lens was used to capture images using 405, 488 and 561 nm laser paths.

Images were analysed using Fiji (ImageJ)[70]. For quantification of mitochondrial proximity to the phagolysosome: Mitotracker positive pixels were quantified at different radial distances from fluorescent zymosan particles (up to 5 µm) and data were expressed as percentage of total mitochondrial localisation. The 3D objects counter[71] was used to calculate approximate mitochondrial volumes, which is based on 65 nm$^2$ per pixel and a 100 nm z-step size, to give voxels of 6500 nm$^3$. 3D models visually fit the data well (Supplementary Movies 1–4).

**Statistical analysis.** Statistical analysis was performed using GraphPad Prism 6 & 7, and the tests used were Student's t-tests, one-way and two-way ANOVAs with post-tests and linear regression, as are indicated in figure legends, all data are assumed normally distributed and were log transformed where appropriate before analysis. *$P < 0.05$, **$P < 0.01$, ***$P < 0.001$. Ingenuity pathway analysis (IPA, Qiagen) was used on metabolomic data sets to identify differences in metabolite pathways. All error bars represent the mean ± the standard error of the mean (SEM unless stated).

**Data availability.** The RNA-seq data discussed in this publication have been deposited in NCBI's Gene Expression Omnibus[72] and are accessible through GEO Series accession number GSE104920. The metabolomics data are attached as Supplementary Data 1–2 and all other data are available from the authors upon request.

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

## Acknowledgements

We would like to thank Laura Coffin for help with luminol assays, Anna Trivett for the purification of human monocytes, Dr. Shurjo Sen and Dr. Fathi Elloumi for performing RNA-seq analysis, Dr. Stephen Lockett of the Optical Microscopy and Analysis Laboratory (OMAL) for development of Fiji macros, Megan Karwan for animal work, and Jessica Walls for preparation of apoptotic thymocytes. This work has been funded in part with federal funds from the National Cancer Institute, National Institutes of Health, Intramural Research Program, USA and the Henry Wellcome Trust, UK (WT103973MA). P.R.T. is funded by the Wellcome Trust (107964/Z/15/Z). D.B.K. is under Contract No. HHSN261200800001E. The content of this article does not necessarily reflect the views or policies of Cardiff University (UK), the Wellcome Trust (UK) Leidos Inc., or the Department of Health and Human Services (USA), nor does mention of trade names, commercial products, or organisations imply endorsement by the U.S. Government.

## Author contributions

L.C.D. conceived and designed the project, conducted and analysed experiments, interpreted the data and wrote the manuscript. D.W.M. facilitated the development and progression of the project, assisted with interpretation of the data, analysed experiments and wrote the manuscript. C.M.R. conducted experiments, assisted with interpretation of data and critically appraised the manuscript. E.M.P. designed mitochondrial experiments, assisted with interpretation of data and critically appraised the manuscript. D.B.K. helped with the design of luminol studies, provided valuable insights on neutrophils and respiratory burst and critically appraised the manuscript. P.R.T. assisted with interpretation of the data, provided valuable knowledge of tissue-resident macrophages, and critically appraised the manuscript.

## Additional information

**Competing interests:** The authors declare that they have no competing financial interests.

