## [Peer Review File · Nature Communications]

Reviewers' comments:

Reviewer #1 (Immunometabolism)(Remarks to the Author):

This paper examine metabolic reprogramming in tissue-resident macrophages, which is an important goal since we have information on metabolic events in isolated macrophages but little recent work on 'real life' macrophages. Some interesting observations are made mainly in relation to a role for Complex II in the generation of ROS following challenge with zymosan. There are however some issues that need to be addressed as follows.

1. OCR is used to measure the respiratory burst in response to zymosan. How do the authors know that the increase in OCR is due to the respiratory burst as opposed to increased respiration? What percentage of the increased OCR is due to the burst versus respiration? This is an important consideration overall as we have to be satisfied that the OCR assay is a reliable indicator of enhanced ROS production. Similarly, it is important to determine how much of the ROS production is coming from the NADPH oxidase system. In essence what would be very useful would be an quantification of the relative roles of respiration, mitochondrial ROS generation and NADPH-oxidase activity here, and then the role of glutamate anaplerosis for each of these. Some of the data are in the paper (eg the authors describe how NOX2 'is required for amplification of the mitochondrial changes', but a more detailed analysis is needed.

2. The authors need to discuss their work in the context of the work of West AP et al Nature (2011) 472, 476-480). That paper finds a clear role for TLR2 in driving mitochondrial ROS and also the movement of mitochondria to the phagolysosome and yet in the current study no evidence for TLRs is found. Why the inconsistency? This should be considered and discussed.

3. Rotenone is clearly having inhibitory effects here (eg Fig 5F, 7E, 8E) and yet the authors mainly emphasise a role for Complex III, for example on p22 where they state 'complex III rather than reported reverse electron transport to complex I'. This needs to be changed as the evidence here also suggests a role for Complex I and the final schematic should include this possibility.

4. In Fig 9 the authors examine cytokine production. Measuring mature IL-1beta without a signal 2 such as ATP or nigericin is odd. The authors should measure IL-1beta mRNA here. also the levels of IL10 are very low which makes that data unreliable. Can the authors get a better magnitude for IL10 production? If not this should be omitted.

Reviewer #2 (Macrophage, inflammation)(Remarks to the Author):

"Tissue-resident macrophages are metabolically poised to engage microbes using tissue-niche fuels"
Nature communications review comment,

General comments,

"Tissue-resident macrophages are metabolically poised to engage microbes using tissue-niche fuels" by Professor Daniel W. McVicar and co-authors, this study implied that metabolites in some tissue is critical for the tissue residential macrophage activation. Especially, peritoneal-resident macrophages utilize glutamate to sustain the respiratory burst for phagocytosis against bacterial infection. This concept may be interesting. Although macrophage characterization is an important area in immunological field, the authors need to more carefully examine the mechanism for macrophage

activation by metabolite. The following specific comments should be addressed.

Specific comments,

In Figure 1 and supplementary figures, heat map of gas chromatography-MS spectrometry in only two types of macrophages were shown. Authors stated that resident macrophages were enriched in tri-carboxylic acid cycle-associated metabolites compared with BM-derived macrophages by M-CSF culture. Why did authors select the "in vitro" BM-derived macrophages as control cell? More careful investigation is needed here. Probably, not only TCA-associated metabolites but also other metabolites or many gene expression pattern was significantly different between the two cell types. To state only TCA-associated metabolites was enriched in residential macrophages, authors should investigate the pattern of metabolites expression in various other macrophage/ monocyte subtypes as well as BM-derived macrophages

In Figure 2, again, more careful analysis need to be done here. Authors showed that heat map of amino acids and fatty acids in peritoneal lavage and serum. Why did authors chose only two samples from lavage and serum? Authors should measure the amino- and fatty- acids in various tissue or location in body. Such prejudiced heatmap analysis data mislead the reader. Thus, authors should analyze the data by using correct control samples.

In Figure 3, Zymosan was used to investigate the respiratory burst in resident macrophage. The authors should test other ligands such as uric acid, alum, dead cells, Pam3CSK4 and R848 for the respiratory burst in the macrophages since peritoneal resident macrophages are activated in response to these ligands.

In Figure 7, the authors stated that the metabolic switch in resident macrophages initiated with Protein Kinase C activation although the switching was occurred p47 independently. Some studies have already indicated that PKC activation occurs downstream of dectin-1, which is receptor for zymosan, and Syk signaling in macrophages. Authors stated that PKC activation was critical for zymosan-induced respiratory burst. Probably the respiratory burst is occurred in various situations. Are other ligands, but not Zymosan, inducing respiratory burst also PKC dependent?

Reviewer #3 (Innate cell programing)(Remarks to the Author):

General Comment: The present study reveals that peritoneal resident macrophages are primed by the tissue-niche fuels to possess specialized metabolic functions that facilitate microbial clearance. This work demonstrates distinct characteristics of peritoneal resident macrophages compared to BMDMs, which have been widely used in macrophage metabolism studies, highlighting physiological relevance. The authors also did extensive work to unveil the mechanisms mediating the metabolic poise of peritoneal macrophages. The experimental design is straightforward and most of the results are solid. The manuscript is well written. Particularly, the schematics are helpful for readers to seize essence of the experiments. The following specific comments are presented to further enhance this manuscript.

Specific Comment (1): Tissue resident macrophages include multiple macrophage subsets mainly based on the tissue distribution, such as peritoneal macrophages, spleen macrophages, Kupffer cells, microglia, etc. The authors may use "peritoneal resident macrophages" instead of "tissue resident macrophages" in the title and text of the paper, since only this macrophage subset was employed in the study. Alternatively, it may be intriguing to determine metabolism features of other tissue resident macrophage subset.

Specific Comment (2): Figure 1 shows that peritoneal macrophages have much higher mitochondrial capacity than BMDMs. The authors may need to demonstrate the morphology of mitochondria that is closely related to mitochondria function.

Specific Comment (3): Figure 1 shows that the elevated mitochondrial capacity of peritoneal

macrophages is primarily dependent on glutamine and fatty acids. However, in Figure 3, the authors tested the influence of glutamine and glucose on respiratory burst. Why was the impact of fatty acids not examined?

Specific Comment (4): The figure legend of Figure 3H is missing.

Specific Comment (5): The data demonstrate that phagocytosis of zymosan is necessary for enhanced respiratory burst in peritoneal resident macrophages. Do microbes, e.g., *S.cer.*, directly trigger the respiratory burst? Is it also dependent on glutaminolysis?

Specific Comment (6): In this study, peritoneal resident macrophages were harvested and then used for the experiments in vitro. The authors may abolish the fuels in the peritoneum first, and then harvest resident macrophages to test whether this treatment can alter the metabolism and bacteria killing capacity of the macrophages.

General reply to all reviewers:

Firstly, we would like to thank the editor and reviewers for their constructive comments. We were pleased the reviewers were very positive about our data and original thesis, and we have now addressed their concerns and comments. Based on their comments, we have performed new experiments and clarified portions of the text, which has substantially improved the manuscript. To incorporate their changes and the new data, we have edited manuscript for length while preserving all original ideas and thesis. In addition, we have changed the order of figures to accommodate inserts, new high-resolution figures have been uploaded and the old and new figure numbers are referred to where appropriate in this response.

Reviewer #1 comments:

'This paper examine metabolic reprogramming in tissue-resident macrophages, which is an important goal since we have information on metabolic events in isolated macrophages but little recent work on 'real life' macrophages. Some interesting observations are made mainly in relation to a role for Complex II in the generation of ROS following challenge with zymosan. There are however some issues that need to be addressed as follows.'

'1. OCR is used to measure the respiratory burst in response to zymosan. How do the authors know that the increase in OCR is due to the respiratory burst as opposed to increased respiration? What percentage of the increased OCR is due to the burst versus respiration? This is an important consideration overall as we have to be satisfied that the OCR assay is a reliable indicator of enhanced ROS production. Similarly, it is important to determine how much of the ROS production is coming from the NADPH oxidase system. In essence what would be very useful would be an quantification of the relative roles of respiration, mitochondrial ROS generation and NADPH-oxidase activity here, and then the role of glutamate aneplerosis for each of these. Some of the data are in the paper (eg the authors describe how NOX2 'is required for amplification of the mitochondrial changes', but a more detailed analysis is needed.'

Firstly, we need to clarify that we do not want to suggest that the enhanced oxygen consumption all directly leads to ROS production. It is an indicator of enhanced mitochondrial function (respiration), which we have shown to correlate with increased superoxide production (Fig.8D) and overall hydrogen peroxide production (Fig.8E). Our data demonstrate a complex interaction between the NADPH oxidase system and mitochondrial function that makes specific assignment of portions of ROS production very challenging. That said, the manuscript provides substantial evidence of the overwhelming involvement of the mitochondria rather than NADPH in the burst response; 1)The hydrogen peroxide production we see is sensitive to mitochondrial complex inhibition (Fig.8E), demonstrating that mitochondrial function is required, 2) we have improved our discussion of potential mechanisms, such as complex III superoxide production and the support of NADPH to fuel NADPH oxidase, 3) when we investigated NADPH levels we found they did not correlate with the cell's ability to increase their oxygen consumption (Fig.8A+C), 4). in the modified manuscript, we provide new supporting evidence to suggest that

NADPH oxidase activity itself is not significantly adding to the enhanced OCR in peritoneal macrophages. We used a specific inhibitor of NADPH oxidase 2, which eliminates neutrophil respiratory oxygen consumption, and found that it has a relatively small effect on peritoneal macrophages (New Fig.5D+E). and 5) we have clarified Fig. 5B by removing the reserve respiratory capacity. These data are already combined to form maximum OCR in Fig.5C. The new look Fig.5B highlights that the non-mitochondrial OCR (some of which will be NADPH oxidase) is smaller than the mitochondrial OCR during respiratory burst, and that there is not a substantial increase in non-mitochondrial OCR from baseline.

Together, these methods corroborate to show that approximately 90% of the OCR increase is not attributed to NADPH oxidase activity itself in peritoneal macrophages. Our pharmacological inhibition of NOX2 is seemingly at odds with the p47^{-/-} macrophages. The former showing little effect while the latter is substantially reduced compared to wild-type macrophages. We attribute this effect to NADPH oxidase activity remaining during attempts to pharmacologically inhibit the enzyme and this has been clarified in the text. Given the different methods used to calculate OCR and superoxide production, we hope the reviewer would agree that calculation of exact ratios of water (OXPHOS) or superoxide production from oxygen in the mitochondria are difficult to ascertain. However, we can calculate the level of proton leak from mitochondria (i.e. electrons failing to make it through to ATP synthase in the electron transport chain), and show that this constitutes a consistent third (31-36%) of mitochondrial OCR that increases proportionally with mitochondrial function after zymosan (Fig.5B). Studies have shown that this type of proton leak results in mitochondrial ROS production¹. Although these numbers can be calculated, we are hesitant to report specific percentages and have rather emphasized the substantial, unexpected involvement of the mitochondria.

‘2. The authors need to discuss their work in the context of the work of West AP et al Nature (2011) 472, 476-480). That paper finds a clear role for TLR2 in driving mitochondrial ROS and also the movement of mitochondria to the phagolysosome and yet in the current study no evidence for TLRs is found. Why the inconsistency? This should be considered and discussed.’

We appreciate the reviewers comment and have now specifically addressed this in the manuscript through experimentation. The reviewer correctly points out that the work of *West et al.* does report a role of TLR receptors in phagolysosome localization, however our data demonstrate that TLR signaling is not required for mitochondrial enhancement after recognition of zymosan. We have now examined the recruitment of mitochondria in TLR2^{-/-} macrophages. Now included in the manuscript is Fig.S9, which shows that mitochondrial localization to phagolysosomes in pResMØ is not dependent on TLR2. It is likely that a range of other receptor signaling is capable of triggering this response, and these mechanisms are likely be different in peritoneal tissue-resident macrophages compared to the immortalized cell lines used in the West et al. study. This means that we now have two key aspects of this response that differ between pResMØ and the cell populations studied by others including West et al. It appears that in pResMØ both the ability to move the mitochondria and the ability to engage complex II are pre-established in vivo and do not require TLR signals *in vitro*. The manuscript has been modified to emphasize this fact.

'3. Rotenone is clearly having inhibitory effects here (eg Fig 5F, 7E, 8E) and yet the authors mainly emphasise a role for Complex III, for example on p22 where they state 'complex III rather than reported reverse electron transport to complex I'. This needs to be changed as the evidence here also suggests a role for Complex I and the final schematic should include this possibility.'

The reviewer is correct, rotenone has inhibitory effects, it is required for respiratory burst, we have tried to make clear that complex I activity doesn't increase during stimulation, and therefore has the same importance before and during respiratory burst. This is shown in the final schematic. We emphasized the role of complex III because its impact increases from baseline (Fig5J, originally 5H) and correlates with the new requirement for complex II. Essentially, in our data all complex I and II activity is dependent on III during respiratory burst, meaning all the electrons are passing forward and there is no evidence of reverse activity (that would still consume oxygen). We made this point in the text, and have now expanded on the explanation in the revision. For clarity to the reviewer, we insert here a representative seahorse plot that was used to calculate the data, it shows that all mitochondrial OCR is dependent on complex III during respiratory burst. We have chosen not to show the data this way in the manuscript as it was more space efficient.

Figure R1: Relative OCR data showing pre- and post-additions of zymosan. Data is representative and from multiple experiments. Pre- and post- treatments are indicated, as is zymosan addition (50ug/ml). Data shown here is n=2-3 per group, error bars are means +/-SEM. The green dotted line represents the level of non-mitochondrial oxygen consumption. Basal OCR is depleted to the same level by antimycin A as rotenone, but respiratory burst OCR is only fully eliminated by antimycin A. The suggestion here is that complex III is required for both complex I and II-fueled OCR.

'4. In Fig 9 the authors examine cytokine production. Measuring mature IL-1beta without a signal 2 such as ATP or nigericin is odd. The authors should measure IL-1beta mRNA here. also the elvels of IL10 are very low which makes that data unreliable. Can the authtors get a better magnitude for IL10 production? If not this should be omitted.'

Our original experiments were aimed to identify specific effects of *Saccharomyces cerevisiae* on IL-1 β production, though we agree adding a second signal such as ATP is useful to look at maximal release (through increased inflammasome activation). Additionally, we originally tried to match our killing assays to cytokine production by using a very small particle number of *Saccharomyces cerevisiae*. Based on the reviewer's comment, we have now performed new experiments, increasing the number of macrophages and *Saccharomyces cerevisiae*; we also used ATP and measured *Il1b* RNA (Replaced Fig.9A, New Fig.9B). The new data are consistent with the original and add strength to these findings. We thank the reviewer for the suggestion.

Reviewer #2 comments:

'General comments,

"Tissue-resident macrophages are metabolically poised to engage microbes using tissue-niche fuels" by Professor Daniel W. McVicar and co-authors, this study implied that metabolites in some tissue is critical for the tissue residential macrophage activation. Especially, peritoneal-resident macrophages utilize glutamate to sustain the respiratory burst for phagocytosis against bacterial infection.

This concept may be interesting. Although macrophage characterization is an important area in immunological field, the authors need to more carefully examine the mechanism for macrophage activation by metabolite. The following specific comments should be addressed.'

'Specific comments,

In Figure 1 and supplementary figures, heat map of gas chromatography-MS spectrometry in only two types of macrophages were shown. Authors stated that resident macrophages were enriched in tri-carboxylic acid cycle-associated metabolites compared with BM-derived macrophages by M-CSF culture. Why did authors select the "in vitro" BM-derived macrophages as control cell? More careful investigation is needed here. Probably, not only TCA-associated metabolites but also other metabolites or many gene expression pattern was significantly different between the two cell types. To state only TCA-associated metabolites was enriched in residential macrophages, authors should investigate the pattern of metabolites expression in various other macrophage/ monocyte subtypes as well as BM-derived macrophages '

The reviewer raises a point we grappled with ourselves; what is the best comparison to make. After consideration, we used bone marrow-derived macrophages as a reference population, because this cell type has been used for the majority of macrophage research, and we felt that any differences may indicate peritoneal tissue-resident macrophage metabolic programming that could be vital for their cell-specific functions in their unique environment. There are caveats to any comparison such as this, explaining why we went to such lengths to show that

our initial metabolomics data were indeed indicative of unique macrophage functions (respiratory burst) which could be supported by their unique environment (glutamate from the peritoneum). Given the reviewer's comment we have clarified our descriptions of the caveats and explained our decision more in the revised text.

Regarding differences other than TCA intermediates, we did identify many other metabolites which were significantly different – these were mentioned briefly in the text and the pattern of metabolite expression shown in Fig.S1. We performed an unbiased IPA analysis (Fig.S1) which told us which pathways were significantly altered, then followed up specifically on this data. In response to the reviewers concern we have included all the raw metabolic data in a table (New TableS1), which could be used as a tool for other researchers to develop new hypotheses. Although some things here could be of additional interest, there are simply too many metabolites and pathways to discuss individually within the scope and word limit of the manuscript.

In response to the reviewer's comment about potential gene expression differences we performed RNA-seq on pResMØ and BMDM (New Fig.S2+3, New TableS2). This work revealed that indeed, as expected the cells had substantially different RNA expression patterns. To our surprise pathway analysis did not show metabolic pathways to be specifically enriched. Interestingly, it appears that in several cases the two macrophage populations express alternate gene isoforms expression and/or opposing expression patterns within pathways. Clearly this supports our contention that these cell populations are vastly different and will have different metabolic programming that could reveal interesting metabolic dependencies of peritoneal tissue-resident macrophages. We thank the reviewer for suggesting this intriguing experiment.

We agree with the reviewer that dissection of niche-metabolic pathway connections in other populations of resident macrophages is very intriguing, though we felt it was critical to initially establish this concept here. We would like to investigate other tissue-resident macrophage populations, but respectfully believe this is out of the scope of our manuscript. We chose peritoneal cells because the vast majority of other tissue-resident macrophages are highly integrated into their respective tissues and invasive digestion techniques would be required to isolate them, which our previous experience would suggest alters cell functions. It will be interesting to see how other tissue-resident macrophages compare to peritoneal macrophages, however such a study will be very technically and resource intensive. We hope that upon publication our work will be used as a model for future studies, as lessons learnt here are likely applicable for other tissue-resident macrophages in their respective tissue niches.

'In Figure 2, again, more careful analysis need to be done here. Authors showed that heat map of amino acids and fatty acids in peritoneal lavage and serum. Why did authors chose only two samples from lavage and serum? Authors should measure the amino- and fatty- acids in various tissue or location in body. Such prejudiced heatmap analysis data mislead the reader. Thus, authors should analyze the data by using correct control samples.'

The reviewer highlights another decision we had to make. Although we attempted to explain this in the original text we have clarified this in the revision. In short, we used serum as a reference nutrient source to identify metabolites because we hypothesized that serum would be rich in metabolic fuels and the peritoneum would be in equilibrium with serum for most metabolites or fuels excepting those of particular need for the resident cell populations. We were surprised at the number of differences we found. Our original manuscript included a summary of the full data in Fig.S5 (originally Fig.S2), and we have now included all the raw data in a table (New TableS3).

Regarding analysis of metabolic niches elsewhere in the body and the idea that macrophages may be specifically metabolically poised there also, we agree. However, as noted above, we felt it was important to establish the concept here first and that a full survey of all the metabolic niches of the body and their corresponding macrophage populations was out of the scope of the study at this time.

'In Figure 3, Zymosan was used to investigate the respiratory burst in resident macrophage. The authors should test other ligands such as uric acid, alum, dead cells, Pam3CSK4 and R848 for the respiratory burst in the macrophages since peritoneal resident macrophages are activated in response to these ligands.

In Figure 7, the authors stated that the metabolic switch in resident macrophages initiated with Protein Kinase C activation although the switching was occurred p47 independently. Some studies have already indicated that PKC activation occurs downstream of dectin-1, which is receptor for zymosan, and Syk signaling in macrophages. Authors stated that PKC activation was critical for zymosan-induced respiratory burst. Probably the respiratory burst is occurred in various situations. Are other ligands, but not Zymosan, inducing respiratory burst also PKC dependent?'

We thank the reviewer for this suggestion. We initially demonstrated no respiratory burst in response to LPS (Fig.3A), and agree with the reviewer that testing other ligands would be interesting. Therefore, we chose a panel of ligands to test, some of which the reviewer recommended, and analyzed them with and without a specific PKC inhibitor. We confirmed that zymosan-induced respiratory burst was dependent on PKC to a similar degree as the PKC agonist PMA (New Fig.7G). Interestingly, none of the other ligands tested were as strong as zymosan at inducing a respiratory burst, though all ligands (apart from LPS) resulted in smaller-magnitude OCR increases, that were glutamine-dependent and somewhat PKC-dependent (New Fig.S6). Interestingly, the magnitude of these bursts was similar to those of zymosan in p47 knockout macrophages. So perhaps these ligands do not effectively activate NADPH oxidase, which is required for the amplification of mitochondrial changes. This has now been discussed within the text.

Reviewer #3's comments:

'General Comment: The present study reveals that peritoneal resident macrophages are primed by the tissue-niche fuels to possess specialized metabolic functions that facilitate microbial clearance. This work demonstrates distinct characteristics of peritoneal resident macrophages compared to BMDMs, which have been widely used in macrophage metabolism studies, highlighting physiological relevance. The authors also did extensive work to unveil the mechanisms mediating the metabolic poise of peritoneal macrophages. The experimental design is straightforward and most of the results are solid. The manuscript is well written. Particularly, the schematics are helpful for readers to seize essence of the experiments. The following specific comments are presented to further enhance this manuscript.'

'Specific Comment (1): Tissue resident macrophages include multiple macrophage subsets mainly based on the tissue distribution, such as peritoneal macrophages, spleen macrophages, Kupffer cells, microglia, etc. The authors may use "peritoneal resident macrophages" instead of "tissue resident macrophages" in the title and text of the paper, since only this macrophage subset was employed in the study. Alternatively, it may be intriguing to determine metabolism features of other tissue resident macrophage subset. '

This is the first study examining tissue-resident macrophages in context with their tissue-niche. However, the reviewer is absolutely correct, we do not want readers to believe that all tissue-resident macrophages will behave in the same manner. As suggested, we have changed the title and mentions within the text to peritoneal tissue-resident macrophages.

Regarding the analysis of additional macrophage populations and their specific niches, again we agree this is an intriguing possibility. However, as mentioned in the response to reviewer #2 above, we felt it critical to use our time and resources to establish this concept here first. We expect that upon publication, the lessons learnt here may also be applicable to other tissue-resident cells in their specialized tissue niches and will foster additional studies.

'Specific Comment (2): Figure 1 shows that peritoneal macrophages have much higher mitochondrial capacity than BMDMs. The authors may need to demonstrate the morphology of mitochondria that is closely related to mitochondria function. '

The reviewer rightly points out that peritoneal tissue-resident macrophages have a higher maximum mitochondrial capacity than bone marrow-derived macrophages. A multitude of factors such as metabolic enzyme expression, differential pathway utilization and morphology could explain these differences. The RNA-expression profiles (New Fig.S2+3, ST2) reveal different citric acid cycle isoform expression which could also impact the maximum capacity. However, we agree that the morphology of mitochondrial networks can also dictate this function. Therefore, we have now quantified mitochondrial volumes of these macrophage populations (New SV1-4, Fig.S4). We find that on average peritoneal tissue-resident macrophages have larger mitochondrial networks (although overall volume matches that of BMDM), which may enhance their function. However, the heterogeneity of mitochondrial

morphology within groups was high. We hypothesize that many factors could affect this, but primarily proliferation rate has been linked to fusion and fission of mitochondria. This has now been included and discussed within the text.

‘Specific Comment (3): Figure 1 shows that the elevated mitochondrial capacity of peritoneal macrophages is primarily dependent on glutamine and fatty acids. However, in Figure 3, the authors tested the influence of glutamine and glucose on respiratory burst. Why was the impact of fatty acids not examined? ‘

We agree that considering a large portion of the maximum capacity is dependent on long-chain fatty acids, it would be useful to know whether they could also support respiratory burst. We have now performed additional experiments to assess the impact of long-chain fatty acids on respiratory burst. The fatty acid inhibitor etomoxir had some small, but non-significant effects on respiratory burst, in contrast to that seen with glutamine deprivation. We conclude that from the two major fuels used to maintain maximum OCR, glutamine is predominantly fueling respiratory burst. We have included this data in a new figure (New Fig.3F).

‘Specific Comment (4): The figure legend of Figure 3H is missing.’

We apologize for this. Following this observation, we have carefully combed through all of the manuscript to check for additional errors. The legend in question has been inserted and the graph type has been changed to match the data presented. Additionally, a few axes titles have been altered (Fig.5G+H, 6D) and error bar descriptions added (New Fig.S7+8 (Old Fig.S3+4)) to clarify the data being presented.

‘Specific Comment (5): The data demonstrate that phagocytosis of zymosan is necessary for enhanced respiratory burst in peritoneal resident macrophages. Do microbes, e.g., *S.cer.*, directly trigger the respiratory burst? Is it also dependent on glutaminolysis? ‘

In response to the reviewers comments we have investigated this and have added new data to the manuscript. As suggested we used *Saccharomyces cerevisiae* and recorded respiratory burst. Interestingly, the magnitude of the respiratory burst observed was lower than that seen with zymosan, perhaps due to different uptake/ receptor mechanisms and ligand exposure, but consistent with our other data, the oxygen consumption recorded was dependent on glutamine. This is discussed within the text and a new figure has been added (New Fig.S6). We thank the reviewer for the suggestion.

‘Specific Comment (6): In this study, peritoneal resident macrophages were harvested and then used for the experiments in vitro. The authors may abolish the fuels in the peritoneum first, and then harvest resident macrophages to test whether this treatment can alter the metabolism and bacteria killing capacity of the macrophages.’

We have given this comment great deal of consideration and discussed how to address it. We agree that being able to examine cell metabolic functions *in vivo* would be fantastic.

Unfortunately, we could think of no way to deplete fuels from the peritoneum that would not be replaced with new metabolites from circulation, or that wouldn't change peritoneal macrophage phenotype through secondary signals from other cell types affected by the removal of fuels. We considered for example, that we could lavage peritoneal fluid, centrifuge out the cells, replace the lavage solution with a small volume of isotonic saline and reinject the cells, though this would drastically alter the cells, and the fuels washed out would likely be replaced by the multitudes of other cells within the peritoneal environment. We additionally considered inhibiting specific metabolic pathways *in vivo*. Though this also has caveats, including the inability to specifically target peritoneal macrophages, meaning multitudes of other peritoneal cells would be affected, which could then in turn cause inflammation or otherwise change peritoneal macrophage phenotype through secondary signals (see figure R2). The reviewer suggests altering the cells *in vivo*, but analyzing *ex-vivo*. We too considered this approach, however intraperitoneal injection of inhibitors will result in unknown concentrations within the peritoneal space as the drug is absorbed systemically, and may also impact other cells. Due to all these caveats, we took the approach of using freshly isolated cells tested *ex vivo*.

Figure R2: Effects of dimethylmalonate *in vivo*. Data shows peritoneal cell populations, expressed as % of total after injection of dimethylmalonate (a complex II inhibitor) or control 2h before an additional injection of zymosan (0.33×10^6 particles). Time after zymosan is indicated, error bars show means +/-SEM. Zymosan injection into the peritoneum results in a quick recruitment of neutrophils and a reduction in tissue-resident macrophages which peaks after a few hours. This relatively mild inflammation starts to resolve by 48h with a drastic reduction in neutrophils and a restoration of tissue-resident macrophages². Malonate treatment is associated with a further decrease in tissue-resident macrophages and increased neutrophil recruitment at 4h. Furthermore, at 48h after injection, neutrophils are still present in the malonate injected mice even without zymosan, additionally tissue-resident macrophages numbers are also reduced

with malonate treatment alone. This demonstrates that malonate itself is promoting an enhancement in- and elongation of- inflammation. Unfortunately, this cannot be attributed to any specific effects on macrophages, as these only number a few million, compared to the vast numbers of other cells within the peritoneum that will have complex II activity.

Editorial comment:

‘In addition, we also suggest using siRNA knockdown of specific enzymes, eg. SDH b subunit, to validate and strengthen the conclusions drawn from the inhibitor studies.’

Similar to consideration of modifying fuels *in vivo*, we considered using siRNA to target metabolic enzymes. However, although we found that complex II inhibition short-term at the baseline has little effect, long term (5h) treatment with these inhibitors results in a dramatic drop in OCR (New Fig.5K). This would be interpreted by some as inhibition of the biology in question, however we feel that this approach is inappropriate and will reflect general toxicity of the agents including secondary effects. In the case of complex II inhibition, we attribute these long-term effects to gradual build-up of TCA intermediates through the loss of low level complex II activity, which would ultimately cause feedback inhibition of the TCA cycle and inhibit the function of complex I (via NAD^+ accumulation). Unfortunately, mRNA knockdown techniques usually take a long time to result in significant reductions in protein (3-10 days), which is dependent on the half-life. As we are trying to assess the function of tissue-resident cells *ex vivo*, with as little alteration as possible, such long-term *in vitro* culture is likely to change the phenotype of the cells to beyond what we examine here. We and others have found that macrophages are metabolically plastic, so prolonged knockdown of complex II subunits would likely change the metabolic profile of the cells reducing their ability to effectively burst, but we subsequently would not be able distinguish the effect of complex II knockdown with other metabolic effects such as reductions in complex I activity, TCA cycle suppression or general stress/ metabolite alterations. We routinely use lentiviral vectors to knockdown other proteins *in vivo*³, however these viruses have the unfortunate side effect of transiently activating the cells and causing peritoneal inflammation, even with scrambled controls. We normally wait for this to resolve, but as noted above it is expected that hitting a fundamental metabolic pathway for the long-term would have unanticipated consequences. Therefore, for this study we chose to investigate immediate effects of specific complex II inhibition with atpenin A5⁴, and validated this approach through the use of two other inhibitors: TTFA and dimethylmalonate, that have been broadly used in the literature^{5,6}. We feel that together with our direct metabolic analysis we have sufficiently tested our hypothesis to the best possible extent.

1. Li X, *et al.* Mitochondrial ROS, uncoupled from ATP synthesis, determine endothelial activation for both physiological recruitment of patrolling cells and pathological recruitment of inflammatory cells. *Can J Physiol Pharmacol* **95**, 247-252 (2017).

2. Davies LC, Rosas M, Smith PJ, Fraser DJ, Jones SA, Taylor PR. A quantifiable proliferative burst of tissue macrophages restores homeostatic macrophage populations after acute inflammation. *European Journal of Immunology* **41(8)**, 2155-2164 (2011).
3. Rosas M, *et al.* The Transcription Factor Gata6 Links Tissue Macrophage Phenotype and Proliferative Renewal. *Science* **344(6184)**, 645-648 (2014).
4. Miyadera H, *et al.* Atpenins, potent and specific inhibitors of mitochondrial complex II (succinate-ubiquinone oxidoreductase). *Proc Natl Acad Sci U S A* **100**, 473-477 (2003).
5. Garaude J, *et al.* Mitochondrial respiratory-chain adaptations in macrophages contribute to antibacterial host defense. *Nat Immunol* **17**, 1037-1045 (2016).
6. Mills EL, *et al.* Succinate Dehydrogenase Supports Metabolic Repurposing of Mitochondria to Drive Inflammatory Macrophages. *Cell* **167**, 457-470 e413 (2016).

REVIEWERS' COMMENTS:

Reviewer #1 (Remarks to the Author):

The authors have dealt adequately with my concerns.

Reviewer #2 (Remarks to the Author):

The revised manuscript represent a tremendous amount of work, and the authors make a very strong statement. This paper is now acceptable.

Reviewer #3 (Remarks to the Author):

The authors have adequately addressed the issues raised, and the manuscript should carry significant information for the field.

REVIEWERS' COMMENTS:

Reviewer #1 (Remarks to the Author):

The authors have dealt adequately with my concerns.

Reviewer #2 (Remarks to the Author):

The revised manuscript represent a tremendous amount of work, and the authors make a very strong statement. This paper is now acceptable.

Reviewer #3 (Remarks to the Author):

The authors have adequately addressed the issues raised, and the manuscript should carry significant information for the field.

We would like to thank all the reviewers for their comments. We believe the previous issues raised by the reviewers have been fully addressed and that their input has greatly improved the manuscript.